# OpenMMEgo: Enhancing Egocentric Understanding for LMMs with Open Weights and Data

**Hao Luo**[1,2]    **Zihao Yue**[3]    **Wanpeng Zhang**[1,2]    **Yicheng Feng**[1,2]

**Sipeng Zheng**[2]    **Deheng Ye**[4]    **Zongqing Lu**[1,2*]

[1] Peking University    [2] BeingBeyond    [3] Remin University of China    [4] Tencent

## Abstract

Recent advances in large multimodal models have significantly advanced video comprehension, yet their performance remains limited in first-person scenarios. The interactive nature of egocentric videos is critical for applications like embodied intelligence, but introduces complex visual contexts that conventional models struggle to capture. To bridge this gap, we introduce OpenMMEgo with innovations across three dimensions: data, model, and training strategy. To provide rich spatiotemporal visual knowledge, we curate a large-scale, high-quality dataset named OME10M, comprising over 8.2M egocentric video QA pairs synthesized from Ego4D series. We also establish OMEBench, a comprehensive benchmark for rigorous egocentric understanding assessment. To alleviate the frequent viewpoint shifts inherent in egocentric videos, we implement semantic-aware visual token compression. Further, a curriculum learning strategy is complemented to foster stable learning across various data complexities. OpenMMEgo consistently improves the performance of LMMs on egocentric benchmarks without sacrificing general video understanding performance. Notably, Qwen2.5-VL tuned with OpenMMEgo substantially outperforms other models of the same size in egocentric video understanding. The data, weights and training code will be put at https://github.com/BeingBeyond/OpenMMEgo.

## 1  Introduction

Despite advance in Large Multimodal Models, showing promise in interpreting well-captured images and videos (Liu et al., 2023, 2024a; Bai et al., 2023), their ability to understand first-person scenarios remains limited (Majumdar et al., 2024; Ye et al., 2025). Unlike third-person recordings, egocentric videos involve the camera wearer as an active participant, with head movements causing frequent camera rotations and viewpoint shifts, posing unique visuospatial challenges. Given their applications in areas like robotics and AR/VR (Yi et al., 2024), empowering LMMs to comprehend first-person videos is urgently needed.

Due to this reason, egocentric video understanding (Grauman et al., 2024; Xu et al., 2025) has gained increasing attention in recent years. Nevertheless, existing efforts remain in their early stages, primarily focusing on specialized tasks like retrieval or grounding (Ye et al., 2025). Some recent works (Ye et al., 2025) convert data from large-scale egocentric datasets (Song et al., 2023) into question-answering formats for instruction tuning. Yet, these datasets are typically either inaccessible or rely on text-only labels of coarse-grained event descriptions. For instance, Lin et al. (2022a); Pramanick et al. (2023) exploit short-term video segments (merely one second), where follow-ups

---

*corresponding author <zongqing.lu@pku.edu.cn>

are also constrained by high-level visual-text pairwise supervision. This overlooks the essence of egocentric scenarios, which demands fine-grained observation to model human activities effectively.

To foster this area, we introduce OpenMMEgo a family of state-of-the-art (SOTA) ego-centric large multimodal models (EgoLMMs), releasing model weights, training data, and learning details. We start from devising a meticulous data curation framework, which advances prior counterparts in two key ways: **i) multi-level visual supervision** and **ii) diverse task types**. First, unlike existing action-centric datasets, we address the scarcity of fine-grained visual cues (e.g., object attributes, scene context, interactions), generate instructional data with both text and RGB-pixel prompts. Specifically, we use textual annotations to synthesize events or action-related data for high-level video semantics. In addition, we extract subtle details (e.g., object motions) from raw frames to offer dense spatial-temporal details for low-level supervision. Second, We incorporate a wide range of egocentric tasks — spanning spatiotemporal, descriptive, and deductive reasoning — to maximize model generalization. Based on this framework, we introduce OME10M, a fine-tuning dataset comprising: 8.2M egocentric video QA pairs covering diverse scenarios and tasks, and over 1M general instruction samples sourced from web videos. This in-box dataset is tailored to equip LMMs with robust egocentric video understanding. Additionally, we include OMEBench, a new challenging benchmark for evaluating egocentric models on complex perception tasks. OMEBench features 4K multiple-choice questions derived from 372 hold-out videos, enabling assessment of EgoLMMs.

We train OpenMMEgo using a standard pretraining paradigm, enhanced by an innovative design motivated by a key observation: *first-person videos exhibit rich spatiotemporal dynamics due to frequent camera shifts. Efficiently capturing these dynamics from dense visual cues is thus crucial.* Prior work shows that visual compression not only reduces computational costs (Kim et al., 2022; Choudhury et al., 2024), but also improves the efficiency of dynamic feature encoding (Shen et al., 2024; Chen et al., 2024b; Cheng et al., 2024; Jin et al., 2024). Inspired by this, we propose **Dual Semantic-aware Token Compression** to derive a compact egocentric representation, DuaSTC comprises two modules: i) *Spatial-redundant Token Merging (STM)*, which aggregates frame-level tokens into higher-level semantic entities, enabling processing in a condensed semantic space; ii) *Temporal-irrelevant Token Pruning (TTP)*, which preserves only motion-salient and semantically critical tokens in the fast frames for local motion understanding, drawing from SlowFast principles (Feichtenhofer et al., 2019; Xu et al., 2024; Huang et al., 2024). TTP mitigates noises from egocentric camera shifts, sharpening focus on dynamic content.

While token compression enhances our model architecture, a key challenge remains: *ensuring training efficacy given the diverse video durations, task types and difficulty levels in our dataset*. To address this, we further devise a **Dual Curriculum Learning Strategy** (Bengio et al., 2009; Wang et al., 2021) with two complementary approaches: i) *Offline Data Curriculum*. We leverage a pre-trained LMM as the reference to assess sample difficulty and categorize them into three tiers. Training progresses from easier to harder examples, allowing gradual adaption egocentric video complexities. ii) *Online Data Dropout*. During training, we dynamically filter samples based on forward loss, excluding the most challenging examples when they exceed the model's current learning capacity. This dual strategy prevents overwhelming the model in early training stages while maximizing learning from appropriate-difficulty samples, improving both efficiency and final performance.

Following this systematic exploration, we conduct extensive evaluations on SOTA video LMMs (Li et al., 2024a; Bai et al., 2025). Our results show that OpenMMEgo consistently enhances egocentric comprehension while maintaining general video understanding capabilities. Notably, it elevates Qwen2.5-VL to achieve new SOTA performance across multiple egocentric benchmarks among models of comparable scale.

## 2 Related Works

**Large Multimodal Models.** Built on Large Language Models (Brown et al., 2020; Ouyang et al., 2022; Touvron et al., 2023) and advnaced vision encoders (Radford et al., 2021; Zhai et al., 2023), LMMs (Fu et al., 2024c; Cheng et al., 2024; Wang et al., 2025) have demonstrated remarkable capabilities in vision-language tasks, driving extensive research. Recent progress focuses on model architecture design (Alayrac et al., 2022; Li et al., 2023, 2024b; Sun et al., 2023), vision-language alignment (Zhu et al., 2023; Liu et al., 2023), high-quality instruction data curation (Gu et al., 2024; Chen et al., 2024a; Wang et al., 2023), and evaluation benchmarks (Fu et al., 2024b; Liu et al., 2024b;

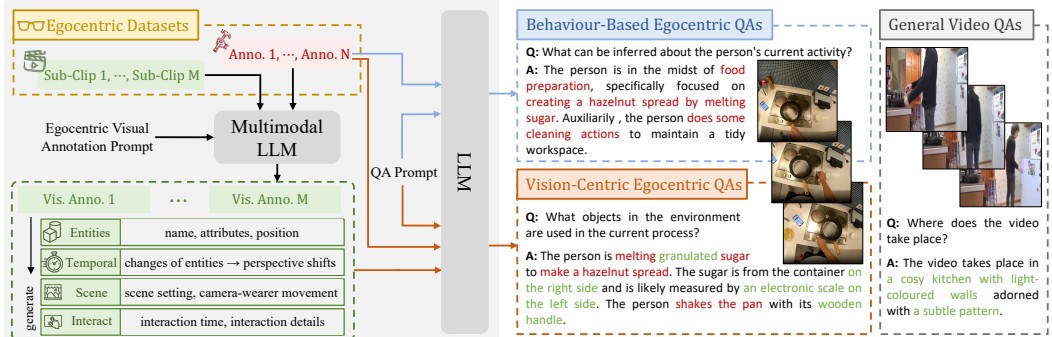

Figure 1: Illustration of OME10M data curation framework (Left) and examples (Right). The action-relevant knowledge and visual-relevant information are systematically processed with (Multimodal) LLMs to synthesize Behavior-Based Egocentric QAs and Vision-Centric Egocentric QAs. OME10M contains both synthesis QAs and web-sourced general video QAs.

Yue et al., 2024a,b). In contrast to third-person view videos, egocentric recordings face unique challenges like scarcity of open-source video data or long visual contexts that exceed model limits. To avoid data scarcity, prior research used synthetic data to train video models (Li et al., 2024a), or adapted image models for video (Xu et al., 2024). Recently, some studies (Chen et al., 2024b; Jin et al., 2024) have explored to manage the extended contexts from overwhelming frame counts by frame sampling techniques (Shen et al., 2024; Ye et al., 2025; Song et al., 2024), efficient frame representation (Li et al., 2024e), and visual token selection or merging algorithms (Shang et al., 2024; Shen et al., 2024; Zhang et al., 2025). In this paper, we combine a large-scale, diverse video dataset with novel strategies for efficient visual token reduction, enabling effective context compression without sacrificing performance on egocentric scenarios.

**Egocentric Video Understanding.** This has emerged as a vital research area due to its real-life application. Early studies like Epic-Kitchens (Damen et al., 2018), Ego4D (Grauman et al., 2022), and Ego-Exo4D (Grauman et al., 2024) focus on proposing new datasets with an increasing scale. These datasets consist of laboratory or daily videos recorded by head-mounted cameras, which provide rich first-person collections through crowd-sourced recordings. For evaluation, question-answering benchmarks such as EgoSchema (Mangalam et al., 2023), EgoPlan (Chen et al., 2023), EgoTaskVQA (Jia et al., 2022), QaEgo4D (Bärmann and Waibel, 2022), and OpenEQA (Majumdar et al., 2024) have been developed alongside specialized models for retrieval and recognition, including EgoCLIP (Lin et al., 2022a), R-VLM (Xu et al., 2023), and Helping Hands (Zhang et al., 2023). While MM-Ego (Ye et al., 2025) represents initial work on egocentric LMMs (EgoLMMs), it remains inaccessible to the public. Our work advances the field through novel data curation, model design, and training recipes, all of which will be open-sourced to support community research efforts.

## 3   OME10M Data Curation

To establish a robust training foundation, we develop a data synthesis framework that delivers multi-level, multi-faceted knowledge specifically designed for egocentric scenarios. Our framework leverages two primary sources: the Ego4D (Grauman et al., 2022) and Ego-Exo4D (Grauman et al., 2024) datasets, which collectively provide 4,900 hours of egocentric videos capturing diverse daily activities (e.g., cooking, cleaning, and repairing), along with brief human annotations describing the events. In addition, we incorporate Ego4D-GoalStep (Song et al., 2023), which offers goal-oriented dense annotations for over 400 hours of video. By combining human annotations with visual information from these videos, we synthesize a comprehensive dataset of **8.2M egocentric video QA pairs**, categorized into two major dimensions: **Behavior-based QAs** and **Vision-centric QAs**.

**Multi-Level Dataset Design.** Our design addresses a critical gap: *while most egocentric instruction tuning datasets rely heavily on high-level human annotations (e.g., actions, goals, events), they overlook low-level visual grounding* — contrasting with training samples for LMMs that focus on basic perceptual signals like object appearances and spatial configurations. This mismatch in supervision granularity may contribute to LMM's inconsistent performance across general and

egocentric video tasks. We posit that that robust egocentric understanding, especially of high-level behavioral semantics (e.g., human intent, procedural goals), requires anchoring in rich low-level visual cues. To bridge this gap, we construct a multi-level dataset combining high-level behavior-based supervision with dense, low-level visual facts, which includes behavior-based QA pairs (human actions, procedural reasoning) and vision-centric QA pairs (granular visual facts: objects, surroundings, and motion patterns). As shown in Figure 1, this joint alternative creates a comprehensive knowledge base. We further augment diversity by incorporating open-source general video QAs to maintain the performance on general video understanding capabilities.

**Visual Details Annotation.** Generating detailed visual annotations for egocentric videos presents unique challenges due to current LMM limitation. While direct annotation using Gemini-1.5-Pro (Team et al., 2024) is possible, we implement a structured pipeline to enhance reliability and coherence. (1) **Pre-annotation**: We first extract core visual facts (objects, environments, fine-grained motions, perspective shifts) from each clip instead of directly generating QA pairs. (2) **Segmented Processing**: To reduce cognitive difficulty, videos are divided into 10-second subclips, each of which is independently annotated with frames sampled at 1FPS. (3) **Multi-step Prompting**: For each subclip, we design egocentric-specific prompts that guide the LMM to output in a structured format: entities, temporal differences, scene setting, and interaction description. Notably, the prompt not only specifies the expected output format but also serves as a form of egocentric task decomposition. It explicitly informs the LMM how each component should be reasoned about and linked: i) tracking identified entities to perceive temporal changes; ii) distinguishing static background from interactive elements to infer perspective shifts and camera motion; and iii) aggregating temporal differences to describe scene layout, body movement, and manipulation details. The reasoning path is directly embedded in the prompt, making annotation a form of structured visual thought, where each step builds upon the previous one. This design improves the reliability and coherence of the annotations. Annotation Examples and the prompt are provided in Appendix B.1.

**Egocentric QAs Synthesis.** With both human and visual annotations in place, we synthesize large-scale QA pairs that support multi-level supervision. Videos are segmented into clips of various lengths (10s, 30s, 1min, 3min, 5min) to cover a spectrum of short- and long-term temporal reasoning. For behavior-based QA, we use human annotations within each video segment as LLM's input, along with prompts covering multiple questioning dimensions, including atomic actions, complex behaviors, and goal-step inference. A total of 3.5M behavior-based QA pairs are synthesized, including 0.9M multiple-choice questions, 0.4M detailed video descriptions, and 2.2M open-ended QA pairs. Similarly, for vision-centric QA, we utilize both human and visual annotations within each segment to guide the synthesis process. Prompts are designed to target diverse visual reasoning perspectives, including spatial-temporal reasoning, self-motion analysis, object interaction, and vision-informed behavioral inference. A total of 4.7M vision-centric QA pairs are synthesized, comprising 1.1M multiple-choice questions, 0.5M detailed video descriptions, and 3.1M open-ended QA pairs. In addition to these synthesized data, we incorporate 1M general video QA pairs from existing large-scale sources such as LLaVA-Video-178k (Li et al., 2024a) and ShareGPT4Video (Chen et al., 2024a), resulting in a final dataset of 9.2M QA pairs, which we name OME10M. To ensure quality, we perform post-hoc filtering based on the predictive loss from an open-source LMM, removing QA pairs with excessively high loss values. Further details and QA examples are provided in Appendix B.1.

**OMEBench.** In addition to training data, we also construct a new benchmark with 372 hold-out videos. Following a similar way of the training data construction, we synthesize 4,000 multiple-choice QA data, comprising a behavior-based subset and a vision-centric subset, for OMEBench. This benchmark, concentrating on both event and visual details, serves as a supplement to existing egocentric benchmarks and can offer useful feedback for model development.

# 4 OpenMMEgo

To enhance EgoLMMs' egocentric comprehension using the rich knowledge provided in OME10M, we target two critical dimensions: *model architecture* and *training strategy*. First, we introduce a dual semantic-aware token compression to handle complex egocentric contents, reducing computational costs while preserving essential visual information (Section 4.1). Second, we employ a dual curriculum learning strategy to optimize training stability and effectiveness (Section 4.2). Together, these innovations pave the way for more robust performance in first-person scenarios.

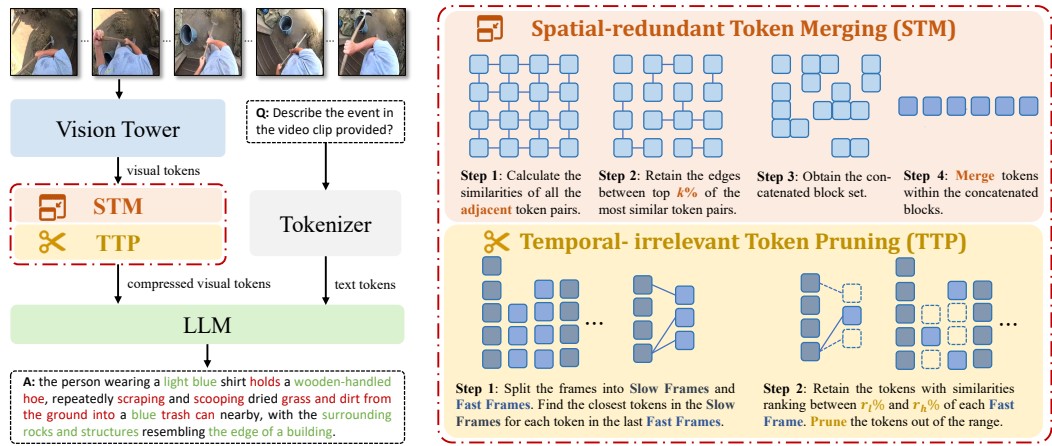

Figure 2: Illustration of dual semantic-aware token compression. The token compression modules are plugged between the vision tower and the LLM. Details of Spatial-redundant Token Merging (STM) and Temporal-irrelevant Token Pruning (TTP) are shown in the right part.

## 4.1 Dual Semantic-aware Token Compression

Egocentric videos exhibit shift viewpoint changes that complicate visual perception — even stationary objects may appear differently across frames. While visual representations vary, the underlying semantics of objects remain consistent throughout the video. This observation reveals two key properties. The intra-frame semantic consistency indicates spatial redundancy which can be reduced to alleviate the comprehension burden. The inter-frame temporal consistency provides valuable hints against viewpoint changes. In this paper, we leverage these properties through: i) **Spatial-redundant Token Merging (STM)** which aggregates tokens representing the identical visual elements into high-level semantic ones; ii) **Temporal-irrelevant Token Pruning (TTP)** which eliminates semantically redundant tokens across frames for efficient temporal modeling. Figure 2 shows the illustration.

**Spatial-redundant Token Merging (STM).** To reduce redundancy in visual tokens, STM aggregates tokens representing identical visual elements. This not only reduces computational cost by decreasing token count, but also enhances temporal reasoning by grouping individual patch tokens into higher-level semantics to correlate these elements across frames. Unlike conventional token merging methods that aggregate the most similar pairs between the two sets after bi-partition (Bolya et al., 2023; Lee et al., 2024; Jin et al., 2024), we exploit the localized nature of visual representations where a specific element is typically represented by its adjacent tokens. Following this, we design STM, a simple yet effective approach to merge adjacent visual tokens into flexible patch islands with varying sizes and shapes. Specifically, STM comprises four steps. **i) Edge Construction**: Build set $E$ containing edges between all pairwise visual tokens that are spatially adjacent. Each edge is weighted by cosine similarity $cos(v_i, v_j)$, where $v_i$ and $v_j$ denotes token features. **ii) Edge Pruning**: Retain only top-$k\%$ weighted edges to form $E^*$, each of which indicates high semantical similarities between adjacent tokens. **iii) Component Detection:** Identify connected components in $E^*$ via a union-find algorithm; **iv) Token Fusion:** Merge tokens within each component into combined semantic tokens.

**Temporal-irrelevant Token Pruning (TTP).** Viewpoint shifts in egocentric videos challenges EgoLMMs to maintain consistent visual understanding across frames. While recognizing cross-frame concept co-occurrence is crucial for viewpoint movement awareness, rapid viewpoint shifts introduce distinct visual changes even in adjacent frames, leading to noises that obscure temporal dynamics. To address this, we propose TTP to simplify the visual context using spatially compressed semantic tokens. Inspired by SlowFast architectures (Feichtenhofer et al., 2019; Xu et al., 2024; Huang et al., 2024), we process frames in two pathways: i) Slow pathway which retains all tokens for spatial detail preservation; ii) Fast pathway which preserves only visual elements that co-occur in slow frames, demonstrating cross-frame invariance to capture temporal dynamics. As Figure 2 illustrates, we calculate cosine similarities between semantic tokens from each slow frame and subsequent fast frames. Tokens exhibiting low semantic similarity are considered view-shift noise, while those with excessive semantic similarity indicate temporal redundancy. TTP prunes both types of tokens, and

retains only semantic tokens within the $[r_l\%, r_h\%]$ similarity percentile range. These remaining tokens maintains crucial visual dynamics cues while eliminating redundant or noisy information.

The synergistic combination of STM and TTP effectively reduces both redundancy and noise in video representations. This dual approach yields two benefits: i) significantly lowering computational demands for long video processing, and ii) enabling the model to better focus on learning meaningful spatiotemporal relationships between semantic entities.

### 4.2 Dual Curriculum Learning Strategy

Unlike previous counterparts, our training dataset OME10M features unique challenges due to its diverse video durations, viewpoint changes, task types and multiple difficulty levels. To address the complexity, we implement this training strategy based on Bengio et al. (2009) combining both offline and online approaches to ensure stable and effective model training.

**Offline Data Curriculum.** We first pre-assess the difficulty of each sample in OME10M, using a pretrained LMM like Qwen2-VL (Wang et al., 2024) as the reference model. Each sample's difficulty is quantized by its forward loss and categorized into three levels `easy`, `medium`, and `hard`, based on loss percentiles. We train OpenMMEgo through three stages with increasing difficulty, each of which involves different data recipes. Due to space limitation, more details can be seen in our Appendix A.2.

**Online Data Dropout.** To further enhance learning efficiency, we introduce this online strategy that dynamically filters challenging in-batch samples during training. For each forward pass, we compute per-sample losses and selectively exclude the hardest samples — those with the highest loss values — from the subsequent backward pass. Formally, given the average batch loss $\bar{l}$ and the loss $l_i$ for the $i$-th sample, the dropout probability is defined as $p_i = \mathrm{Clip}\{\alpha \cdot (l_i - \bar{l}), 0, 1\}$. By iteratively pruning overly difficult samples, this online strategy ensures the model prioritizes samples with higher learnability at its current training state, fostering more efficient optimization.

## 5 Experiments

### 5.1 Experimental Setup

**Implemented Details.** To evaluate the effectiveness of OpenMMEgo, we apply it to two state-of-the-art 7B video MLLMs, LLaVA-Video (Zhang et al., 2024b) , and Qwen2.5-VL (Bai et al., 2025), to train two variants of OpenMMEgo. For visual token compression, we set $k = 35$ for STM and $r_l = 35$ and $r_h = 95$ for TTP. To enhance offline difficulty estimation in dual curriculum learning, we leverage two additional models, `LLaVA-OV-7B` (Li et al., 2024a) and `Qwen2-VL-7B` (Wang et al., 2024), and compute a combined loss from both to derive the final difficulty score for each training sample. The hyperparameter $\alpha$ for online in-batch data dropout is set to 0.3. In our implementation, each video is processed as up to 192 frames (resized $384 \times 384$), with a maximum visual token context length of $N = 13,440$. Following the training framework of LLaVA-Next [2], we train both variants for 1 epoch with a global batch size of 128 across 128 NVIDIA A800 GPUs.

**Baselines.** To provide a comprehensive evaluation, we compare our models with a series of state-of-the-art open-sourced video MLLMs of similar scale. Most relevant is MM-EGO (Ye et al., 2025), an EgoLMM built upon LLaVA-OneVision (Li et al., 2024a). Although MM-EGO is not yet publicly available, we include its reported performance for comparison.

### 5.2 Main Results

We assess OpenMMEgo from two aspects: i) its effectiveness in enhancing egocentric video understanding, and ii) its potential trade-offs in general video understanding capabilities after egocentric adaptation. To systematically evaluate these aspects, we conduct comprehensive benchmarking against SoTA and open-sourced video LMMs across representative question-answering tasks.

**Egocentric Video Understanding.** To thoroughly evaluate improvements on this aspect, we assess OpenMMEgo on five multiple-choice QA benchmarks focusing on egocentric scenarios. We use EgoSchema (Mangalam et al., 2023) for comprehensive evaluation, while the validation set

---

[2] https://github.com/LLaVA-VL/LLaVA-NeXT

Table 1: Comparisons with state-of-the-art LMMs on egocentric video understanding tasks. We evaluate accuracy (%) across 5 multiple-choice QA benchmark, with the best results highlighted in **bold**. For egocentric-oriented LMMs, we report the performance gap relative to their initialized LMMs in parentheses, where positive improvement are underlined.

| | EgoSchema | EgoPlan | QAEgo4D | EgoTaskVQA | OMEBench (beh.) | OMEBench (vis.) |
|---|---|---|---|---|---|---|
| # General Video MLLMs | | | | | | |
| Chat-UniVi (Jin et al., 2024) | 45.1 | 22.1 | 36.2 | 35.4 | 25.1 | 23.7 |
| VideoLLaMA2 (Cheng et al., 2024) | 51.7 | 28.9 | 47.4 | 44.0 | 31.2 | 34.3 |
| VideoChat2 (Li et al., 2024c) | 54.4 | 26.2 | 43.6 | 45.2 | 32.5 | 35.8 |
| LLaVA-OneVision (Li et al., 2024a) | 60.1 | 32.4 | 55.2 | 53.6 | 46.7 | 40.5 |
| Qwen2-VL (Wang et al., 2024) | 66.7 | 39.2 | 55.4 | 54.1 | 48.1 | 45.3 |
| LLaVA-Video (Zhang et al., 2024b) | 57.3 | 37.5 | 52.2 | 50.3 | 47.5 | 44.3 |
| InternVL2.5 (Chen et al., 2024c) | 51.5 | 32.1 | 51.4 | 48.5 | 44.3 | 40.2 |
| Qwen2.5-VL (Bai et al., 2025) | 65.0 | 45.2 | 59.4 | 53.7 | 55.3 | 49.4 |
| VideoChat-Flash (Li et al., 2024d) | 64.3 | 40.8 | 62.4 | 54.8 | 57.1 | 54.5 |
| # Egocentric-Oriented MLLMs | | | | | | |
| MM-EGO (Ye et al., 2025) | 69.0 (+8.9) | - | - | - | - | - |
| **OpenMMEgo (LLaVA-Video)** | 65.8 (+8.5) | 46.7 (+9.2) | 62.0 (+9.8) | 55.4 (+5.1) | 64.4 (+15.0) | 59.3 (+15.7) |
| **OpenMMEgo (Qwen2.5-VL)** | **69.3** (+4.3) | **50.2** (+5.0) | **65.6** (+6.2) | **56.2** (+2.5) | **65.7** (+10.4) | **63.2** (+13.8) |

of EgoPlan (Chen et al., 2023) and the closed test set of QAEgo4D (Di and Xie, 2024) specialize in assessing goal-step reasoning and episodic memory, respectively. To evaluate generalization across video sources beyond Ego4D, we evaluate on the indirect test set of EgoTaskVQA (Jia et al., 2022) using LEMMA (Jia et al., 2020) videos. Additionally, our OMEBench provides in-domain evaluations with two splits: behavior-based QAs (OMEBench$_{(beh.)}$) and vision-centric QAs (OMEBench$_{(vis.)}$), aligned with OME10M's data synthesis pipeline. We conduct zero-shot evaluations on all benchmarks except OMEBench. As shown in Table 1, OpenMMEgo significantly outperforms its base models across all benchmarks, demonstrating the ability to capture egocentric video knowledge from OME10M and generalize effectively. Notably, The Qwen2.5-VL variant of our model surpasses all other LMMs of comparable size.

**General Video Understanding.** We further investigate how introducing egocentric video knowledge affects general video understanding. To this end, we evaluate OpenMMEgo on the following benchmarks: Video-MME (Fu et al., 2024a) (w/o subscripts), MVBench (Li et al., 2024c), and PerceptionTest (Pătrăucean et al., 2023), which measure models' capabilities in video perception, reasoning, and related aspects. The results on these benchmarks are presented in Table 2. When compared with the base model before adaptation, OpenMMEgo maintains stable performance on general benchmarks, with only minor fluctuations — slight drops on MVBench for LLaVA-Video and on Video-MME for Qwen2.5-

Table 2: Comparisons with state-of-the-art LMMs on general video understanding tasks. We provide accuracy (%) and best results highlighted in **bold** as well.

| | Video-MME | MVBench | PerceptionTest |
|---|---|---|---|
| # General Video MLLMs | | | |
| Chat-UniVi | 40.6 | - | - |
| VideoLLaMA2 | 47.9 | 54.6 | 51.4 |
| VideoChat2 | 39.5 | 60.4 | 47.3 |
| LLaVA-OneVision | 58.3 | 56.7 | 57.1 |
| Qwen2-VL | 63.3 | 67.0 | 62.3 |
| LLaVA-Video | 63.3 | 70.8 | 67.9 |
| InternVL2.5 | 64.2 | 72.0 | 68.2 |
| Qwen2.5-VL | 65.1 | 69.6 | 70.5 |
| VideoChat-Flash | **65.3** | **74.0** | **76.2** |
| # Egocentric MLLMs | | | |
| MM-EGO | 57.0 (-1.3) | - | - |
| **OpenMMEgo (LLaVA-Video)** | 63.4 (+0.1) | 70.6 (-0.2) | 68.1 (+0.2) |
| **OpenMMEgo (Qwen2.5-VL)** | 65.0 (-0.1) | 70.8 (+1.2) | 71.2 (+0.7) |

VL, alongside positive gains elsewhere. We attribute this to our well-designed training strategies based on dual curriculum learning. Notably, MM-EGO, initialized from LLaVA-OneVision, exhibits a more pronounced performance decline on Video-MME compared to its base model. This contrast underscores the advantage of OpenMMEgo's plug-in adaptation, which minimizes interference with general video understanding. Moreover, OpenMMEgo-Qwen2.5-VL achieves performance on par with state-of-the-art LMMs. In summary, our work effectively enhances egocentric video understanding without compromising — sometimes even improving general video comprehension.

**Additional Baselines and Benchmarks.** To complement the main results, we provide additional experiments with new baselines and benchmarks in Appendix D.1. The new baselines include

Table 3: The results of ablation experiments on data components and curriculum learning of Open-MMEgo with OpenMMEgo-Qwen2.5-VL. We report the accuracy (%) of multiple-choice QAs on 5 egocentric video benchmarks, and the average accuracy (%) on 3 general video benchmarks (Video-MME, MVBench, and PerceptionTest) is reported under 'Gen. Ben.'. The performance gap of each ablation compared to OpenMMEgo-Qwen2.5-VL is shown in parentheses.

| | EgoSchema | EgoPlan | QAEgo4D | EgoTaskVQA | OMEBench (beh.) | OMEBench (vis.) | Gen. Ben. |
|---|---|---|---|---|---|---|---|
| # Ablations on Data Components | | | | | | | |
| w/o Behavior-Based QAs | 66.1 (-3.2) | 48.1 (-2.1) | 64.4 (-1.2) | 54.8 (-1.4) | 58.6 (-7.1) | 59.0 (-4.2) | 68.5 (-0.5) |
| w/o Vision-Centric QAs | 66.5 (-2.8) | 45.1 (-5.1) | 62.4 (-3.2) | 55.4 (-0.8) | 59.4 (-6.3) | 52.1 (-11.1) | 68.3 (-0.7) |
| w/o General Video QAs | 67.6 (-1.7) | 49.4 (-0.8) | 65.0 (-0.6) | 55.9 (-0.3) | 63.6 (-2.1) | 60.2 (-3.0) | 67.9 (-1.1) |
| # Ablations on Curriculum Learning | | | | | | | |
| w/o Online Curriculum | 67.9 (-1.4) | 50.3 (+0.1) | 63.6 (-2.0) | 54.9 (-1.3) | 63.7 (-2.0) | 60.1 (-3.1) | 68.2 (-0.9) |
| w/o Offline Curriculum | 67.2 (-2.1) | 48.7 (-1.5) | 64.8 (-0.8) | 53.5 (-2.7) | 60.6 (-5.1) | 59.0 (-4.2) | 69.3 (+0.3) |

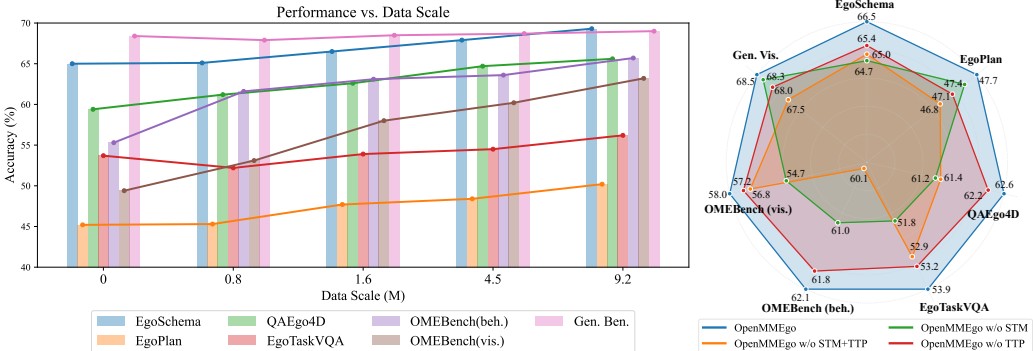

Figure 3: The performance of OpenMMEgo-Qwen2.5-VL on benchmarks across training dataset scales (left) and token compression variants (right). In the left chart, each colored line corresponds to the trend of one benchmark. In the right radar chart, we report performance of OpenMMEgo variants trained with 1.6M QA pairs sampled from the dataset due to the resource limit.

closed-source and synthetic-data models for fairer comparison on egocentric tasks, while the new benchmarks cover datasets beyond Ego4D to assess broader generalization.

## 5.3 Ablation Study

We conduct comprehensive ablation experiments to investigate the impact of each part in OpenM-MEgo on egocentric video understanding and general video understanding. The ablation experiments are performed with OpenMMEgo-Qwen2.5-VL, and the results are demonstrated in Table 3 and Figure 3. We evaluate the performance of each ablation on the same benchmarks as in Section 5.2.

**Q1: Whether all of the components in OME10M are essential and effective?** The results in Table 3 show that excluding any sub-dataset leads to a noticeable drop in performance across the benchmarks, underscoring the significance of all three components in OME10M. Notably, the removal of vision-centric egocentric QAs results in the most pronounced decline, nearly halving the improvement observed on the in-domain OMEBench and even yielding performance inferior to the base model Qwen2.5-VL on EgoPlan. Featured with more vision-centric facts, vision-centric egocentric QAs may act as a pivotal bridge, assisting models to aggregate fundamental visual elements into higher-order, behavior-oriented knowledge within the egocentric perspective.

**Q2: Whether the LMMs benefit consistently from scaling up the training dataset ?** Given that all the synthesized data originates from the Ego4D dataset family and constitutes a major portion of OME10M, we evaluate the performance of the models trained with different data scales. As shown in Figure 3, the model exhibits steady performance gains across all benchmarks as the training data scale increases. Notably, the model does not suffer from overfitting as the data scale grows, which reflects the effectiveness of the data synthesis. Through a multi-level synthesis framework, OpenMMEgo provides rich and diverse visual supervision to continuously empower the LMMs.

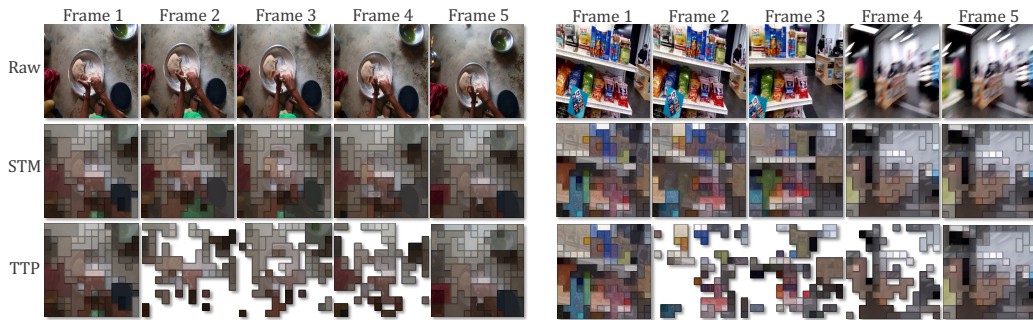

Figure 4: Visualization of the token compression process on two video clips with OpenMMEgo-LLaVA-Video.

**Q3: Whether both of the curriculum learning strategies facilitate better data utilization?** We exclude one curriculum strategy at a time and report the performance in Table 3. The results show that relying on a singular approach leads to a degradation in overall performance, which implies that the two curriculum learning strategies function complementarily. Thus, the improvements achieved by OpenMMEgo in egocentric video understanding stem not merely from increased data volume but from the effective utilization of data diversity and richness.

**Q4: Whether the visual token compression modules contribute to performance?** We conduct ablation experiments on the two visual token compression modules in OpenMMEgo. Since different compression strategies support varying maximum numbers of video frames, we evaluate each configuration under its maximum allowable frame length, meaning that the comparison is made under approximately equal total visual token budgets. As shown in Figure 3, introducing the STM module into the uncompressed model leads to consistent performance gains across benchmarks. Further adding TTP on top of STM brings additional improvements. However, directly applying TTP without STM yields mixed results, with 3 of the benchmarks improving and the others declining. As TTP is designed based on semantic entity grouping, which may not hold in the absence of STM and thus introduces noise. These findings suggest that blindly increasing the number of input video frames does not necessarily improve performance. Instead, performance gains arise from selective attention to critical visual information. In this regard, the combination of STM and TTP proves effective.

We also report additional analyses on model design in Appendix D.2. These include comparisons with alternative strategies for token compression and curriculum learning, as well as hyperparameter ablations of key components.

## 5.4 Token Compression Cost vs. Benefit

We further investigate the computational trade-off of our compression design. To ensure fairness, we normalize the inference time of Qwen2.5-VL without any compression of OpenMMEgo to 1.0 and report relative cost for other variants. As shown in Table 4, applying STM or TTP alone moderately increases the maximum supported frames (from 64 to 106 or 91), with inference time rising to 1.31× and 1.19× respectively. When combining STM+TTP, OpenMMEgo processes up to 192 frames, which is nearly *three times* the base model, while the inference cost is only 1.47×. This demonstrates that our joint compression strategy yields a highly favorable cost–benefit tradeoff, enabling longer egocentric sequences to be handled within practical budgets.

Table 4: Computation overhead and maximum supported frames under different compression settings. Inference time is normalized to the base model without any compression (w/o STM+TTP), set as 1.0.

| Variant | Max Frames | Normalized Inference Cost |
|---|---|---|
| Qwen2.5-VL | 64 | 1.00 |
| Qwen2.5-VL w. TTP | 106 | 1.31 |
| Qwen2.5-VL w. STM | 91 | 1.19 |
| **OpenMMEgo (STM+TTP)** | **192** | **1.47** |

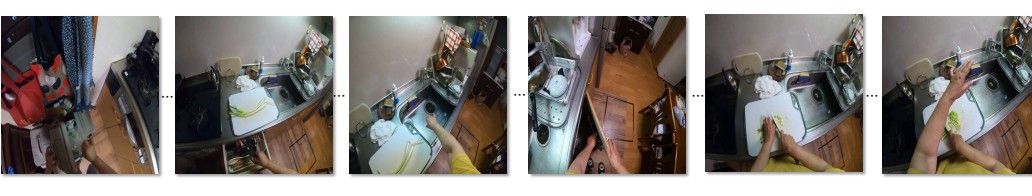

👤 **Q1:** Where are the bags and piece of clothing hanging?

♟ **OpenMMEgo:** The bags and clothing are hanging on the wall above the trash cans.

👤 **Q3:** Why does the camera-wearer open the cabinet under the counter?

♟ **OpenMMEgo:** To retrieve a knife for cutting the spring onions.

👤 **Q2:** What does the camera-wearer first do after throwing away an empty plastic bag?

♟ **OpenMMEgo:** The camera-wearer first puts the scissors on the counter back into the drawer below.

👤 **Q4:** Which hand does the camera wearer use to discard the small piece of spring onion?

♟ **OpenMMEgo:** The camera wearer discards the spring onion piece with the left hand.

Figure 5: QA examples of OpenMMEgo-Qwen2.5-VL on a two-minute egocentric video clip. The model can answer these questions regarding environmental understanding, temporal reasoning, goal-step reasoning, and action detail perception.

## 5.5 Case Study

In this subsection, we pose some specific cases for analysis.

**Token Compression Visualization.** We visualize the token compression process in OpenMMEgo on two video clips with OpenMMEgo-LLaVA-Video in Figure 4. In the STM, tokens corresponding to patches with similar semantics are merged together. In the TTP, tokens related to local motion on fast frames are retained, such as the tokens related to the hand in the top case. Even in the bottom case, where there is a significant perspective shift and even noise, TTP still retains some tokens that can be used for motion understanding, such as the items on the shelf and the blurred checkout counter by the door. This reflects that the semantic-aware visual token compression module in OpenMMEgo effectively compresses the visual context across different scenes.

**QA Examples of OpenMMEgo.** For a better understanding of our model capabilities, we present examples of generated results from OpenMMEgo-Qwen2.5-VL. As shown in Figure 5, we propose 4 questions over a two-minute egocentric video clip, which is sourced from Ego4D and is not included in the training data. The questions are designed to examine the model's abilities in environmental understanding, temporal reasoning, goal-step reasoning, and action detail perception. Answering these questions requires capturing basic visual information, perceiving actions from the video, and performing some reasoning. OpenMMEgo-Qwen2.5-VL can answer these questions well, demonstrating its capabilities in egocentric video understanding.

## 6 Conclusion

In this paper, we present OpenMMEgo to enhance LMMs in egocentric video understanding from data, model, and training aspects. We construct a large-scale dataset, OME10M, containing 8.2M egocentric video QA pairs synthesized from Ego4D series. We also introduce a benchmark, OMEBench, for comprehensive evaluation of EgoLMMs. To alleviate the frequent perspective shifts in egocentric videos, we implement semantic-aware visual token compression and employ a curriculum learning approach to foster stable learning across various data complexities. Our experiments on 7B size LMMs demonstrate that OpenMMEgo significantly improves the performance LMMs on egocentric benchmarks without sacrificing general video understanding performance. Notably, Qwen-2.5-VL tuned with OpenMMEgo outperforms other models of the similar size in egocentric video understanding.

**Acknowledgments**

This work was supported by NSFC in part under Grant 62450001 and 62476008. The authors would like to thank the anonymous reviewers for their valuable comments and advice.

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

# A  Implementation Details

## A.1  Training Details

We follow the training framework of LLaVA-Next [3], using a learning rate of $1e^{-5}$ and a warmup ratio of 0.03 throughout the entire instruction tuning process. As for the based models, we use the version of `LLaVA-Video-7B-Qwen2` [4] and `Qwen2.5-VL-7B-Instruct` [5] respectively. The training process is conducted on $128 \times$ A100 GPUs with the hyperparameters listed in Table 5. With at most 5 QA pairs on the same video clip integrated in one single data point, the whole training process takes 31 hours for 1 epoch.

Table 5: Training hyperparameters for instruction tuning.

| Hyperparameter | Value |
|---|---|
| Global Batch Size | 128 |
| Frame Number | 192 |
| Input Resolution | 384 |
| Learning Rate | $1e^{-5}$ |
| Weight Decay | 0 |
| Warmup Ratio | 0.03 |
| Learning Rate Scheduler | cosine |
| Numerical Precision | bfloat16 |
| Epochs | 1 |
| Max Sequence Length | 32768 |
| Max Visual Context Length | 13440 |

## A.2  Curriculum Details

In our offline curriculum learning, we quantify the difficulty of each data via the loss value on capable LMMs. To refine our dataset, we first exclude data points exhibiting excessively high loss values, with a threshold set at 3.0 in our implementation. Similarly, for benchmark construction, we select the 4,000 QA pairs with the highest loss values, constrained to those below 3.0. According to the loss values, the filtered dataset is divided into three subsets **in equal size**, `easy`, `medium`, and `hard`. We then organise the datasets for a three-stage training, with the overall difficulty increasing as the stages progress. The specific recipe for each stage is presented in Table 6.

Additionally, the distribution of difficulty scores across the three splits of OME10M is depicted in Figure 6. According to our difficulty metric, all three splits exhibit a long-tail distribution. Notably, the general video QAs generally exhibit lower difficulty compared to egocentric QAs. Within the egocentric data, the vision-centric egocentric QAs exhibit relatively lower difficulty, with a more even distribution across various difficulty levels, in contrast to the behavior-based egocentric QAs. This distribution pattern suggests that Vision-Centric Egocentric QAs are more amenable to learning, as they incorporate fundamental visual facts, thereby partially bridging the gap between Egocentric QA and General Video QA.

Table 6: Data partition for each training stage.

|  | Easy | Medium | Hard |
|---|---|---|---|
| Stage-1 | 60% | 30% | 10% |
| Stage-2 | 35% | 50% | 15% |
| Stage-3 | 5% | 20% | 75% |

---

[3] https://github.com/LLaVA-VL/LLaVA-NeXT

[4] https://huggingface.co/lmms-lab/LLaVA-Video-7B-Qwen2

[5] https://huggingface.co/Qwen/Qwen2.5-VL-7B-Instruct

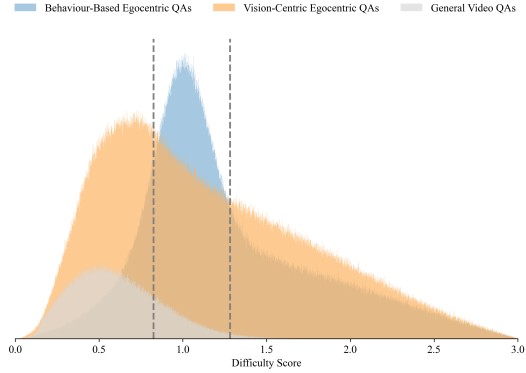

Figure 6: Distribution of the difficulty scores for offline curriculum learning over the 3 splits of OME10M. Dashed lines indicate the tri-quantile points of the difficulty scores used to split the dataset into `easy`, `medium`, and `hard` subsets.

## A.3 Egocentric Benchmarks

In addition to our OMEBench, we also conduct experiments on several open-source egocentric benchmarks for a thorough evaluation in egocentric video understanding.

**EgoSchema.** A long-form egocentric video QA benchmark sourced from Ego4D, designed to assess temporal reasoning over 3-minute clips, with a focus on high-level activity understanding through multiple-choice questions.

**EgoPlan.** A goal-oriented benchmark that evaluates planning ability in egocentric contexts by predicting the next plausible action in a task sequence. We adopt the validation set for evaluation.

**QAEgo4D.** A egocentric benchmark derived from Ego4D, targeting episodic memory in first-person videos. Models are required to select concise, factual answers from a fixed vocabulary. We use the closed-set version of QAEgo4D for evaluation.

**EgoTaskQA.** A diagnostic benchmark based on the LEMMA dataset, designed to assess goal-directed reasoning through four types of questions: descriptive, predictive, explanatory, and counterfactual. We use the indirect test set to increase evaluation difficulty. To enable consistent evaluation, we convert the open-ended questions into a multiple-choice format by prompting `Gemini-1.5-pro` to generate 2–4 distractor options based on the correct answer.

# B Dataset Details

## B.1 Data Synthesis Framework

In this section, we describe the details of the data synthesis framework. First, we segment the videos into clips spanning the range of 0–5 minutes, specifically 10s, 30s, 1min, 3min, and 5min. These clips serve as the QA targets of OME10M. We utilize the `Gemini-1.5-pro` to generate QA pairs for the clips in two manners, respectively producing behavior-based QAs and vision-centric QAs.

For behavior-based data synthesis, we provide all the human annotations associated with each video clip to the `Gemini-1.5-pro` for QA generation. To enhance the quality of the generated questions, we predefine several question types, which can be categorized into three main levels: (1) atomic actions, including action recognition, interactive object grounding, action temporal reasoning, and action counting; (2) chunked behaviors, where atomic actions are aggregated into descriptive behavior paragraphs; and (3) goal-step reasoning, encompassing goal analysis, next-step prediction, and precondition identification. The human annotations relevant to each video clip are iteratively provided to `Gemini-1.5-pro` multiple times, prompting separate data synthesis from these distinct levels. Besides the predefined aspects, we also prompt `Gemini-1.5-pro` to generate free-form QAs based on the information. The prompt of behavior-based data synthesis is provided in Figure 7 and Figure 10.

For vision-centric data generation, we first segment the original videos with 10-second intervals, which are then densely annotated with spatial-temporal information relevant to vision-centric analysis. Specifically, each short clip is frame-extracted at 1 fps, and both the extracted frames and corresponding human annotations are input into `Gemini-1.5-pro` for vision-centric annotation. In the system prompt, we include necessary step-by-step reasoning guidance, which covers the following:

1. Analyze the spatial location, shape, color, and size of all objects in each frame.

2. Compare adjacent frames, identifying the changes in the background objects to annotate the perspective shift between frames. Infer motion information, such as whether the camera wearer is moving or turning in a specific direction, based on perspective shifts.

3. Describe the scene in detail according to the spatial relationships among the objects.

4. Annotate the action descriptions to specify the interactive objects and their interaction details.

The prompt used for visual annotation synthesis is provided in Figure 8.

Based on the four-dimension vision annotations at the granularity of 10-second segments, we generate vision-centric QA data in a similar manner. We also specify several question directions for generating vision-centric QAs: (1) spatial-temporal reasoning, focusing on object features, object relationships, and scene descriptions; (2) self-motion analysis, mainly questioning about the motion of the camera wearer reflected by perspective changes; (3) object interaction, querying the features and details of the interacting objects; and (4) vision-injected behavior, which incorporates the question types in behavior-based QAs but requires the model to integrate annotated visual information in the synthesis QA. The vision-centric annotations, together with corresponding human annotations, are fed to `Gemini-1.5-pro` in the same manner as the behavior-based data synthesis. The prompt of vision-centric data synthesis is provided in Figure 9 and Figure 10.

## B.2 OME10M and OMEBench Examples

We provide more examples of the synthesized QA pairs in OME10M. As shown in Figure 11 (Left), the generated QAs cover a wide range of topics in egocentric videos, including movement analysis, causal reasoning, environmental understanding, next-step prediction, detailed action description. Additionally, we also provide examples from OMEBenchtogether with the corresponding answers from OpenMMEgo-Qwen2.5-VL and Qwen2.5-VL. As shown in Figure 11 (Right), OpenMMEgo-Qwen2.5-VL is able to answer the questions correctly while Qwen2.5-VL fails to do so. These examples involve the capability of environmental understanding, action recognition, temporal grounding, and object motion perception. The outperformance of OpenMMEgo-Qwen2.5-VL over Qwen2.5-VL in these examples indicates the effectiveness of our method in enhancing the egocentric video understanding capability of MLLMs.

## C  Additional Related Works

**Visual Token Merging.** Visual token merging has demonstrated significant effectiveness in improving representation efficiency while reducing computational costs, initially in image-classification tasks. Key techniques include similarity-based merging (Bolya et al., 2023), learned merge-ratio (Feng and Zhang, 2023), learned threshold-based merging and pruning (Bonnaerens and Dambre, 2023), decoupled embedding modules (Lee and Hong, 2024), and depth-aware spatial token aggregation (Huang et al., 2025). These approaches have since been extended to video inputs, with innovations such as learnable spatiotemporal merging (Lee et al., 2024), spatial-temporal token selection (Wang et al., 2022), and temporal interpolation (Zhang et al., 2024a). Similarly, in large multimodal models (LMMs), token compression has been adopted to improve inference efficiency and handle long contexts, including methods based on static pixel token reduction (Zhang et al., 2025), cross-modal querying reduction (Shen et al., 2024), and hierarchical compression (Li et al., 2024d). Building on these advances, we propose a method specifically designed for egocentric videos, which present unique challenges due to dense visual cues and frequent camera motion. We introduce a dual-branch compression strategy that restricts spatial token merging to adjacent tokens for preserving semantic entity boundaries, and performs temporal pruning over entity-aligned tokens in a SlowFast-inspired view to retain motion-salient information.

**Input & Task**
You are tasked with generating high-quality. | Question-Answer (QA) pairs | | Multiple-Choice Questions (MCQs) | to evaluate participants' understanding of egocentric videos. Instead of raw video input, you are provided human annotations of the atomic actions are also provided to you. Some marks are used in the narrations: '#C' for the camera wearer, '#not sure' for the uncertain information, '#summary' for the summary of the narration sequence, '#O' for other action conductors.
Your task is to generate question-answer pairs that probe the participant's comprehension of visual, temporal, spatial, causal, motor, and interactional aspects of egocentric video, based on the human annotations.

> **Each question must have one correct answer and three plausible distractors.** The distractors should be **confusing but incorrect**, grounded in elements that appear in the video but do not satisfy the question, or artificial elements that could reasonably be inferred but are wrong.

**Guidelines:**
1. **Question Scope**: You should generate questions about **{question_scope}**. You should generate appropriate questions in these angles or their composition.
2. **Question-Answer Quality:**

> The generated questions should vary in complexity and scope. Across the generated QA pairs, ensure coverage of diverse question angles within the assigned dimension. Avoid producing repetitive patterns or overly similar question formulations. Some questions should target fine-grained recognition and result in short, factual answers. Others should require reasoning across multiple behaviors, resulting in longer, integrative answers. Both short-form and long-form questions must be present in each generation batch. For the diversity of questions, only part of the information should be involved in each QA pair. Though the answers may be long, the questions should be short and concise to hit the nail on the head.

> The generated questions should vary in complexity and scope. Across the generated MCQs, ensure coverage of diverse question angles within the assigned dimension. Avoid producing repetitive patterns or overly similar question formulations. Some questions should target fine-grained recognition and others should require reasoning across multiple behaviors. For the distractors:
> a) Distractors must be plausible in the scene or event.
> b) They should differ subtly from the correct answer, e.g., similar object, wrong time, incorrect goal.
> c) Avoid obviously false or out-of-scope distractors.

3. **No Answer Leakage:** The answer must not be directly inferable from the question text alone. Avoid phrasing that embeds hints, reuses descriptive terms from annotations, or overly specifies entities or outcomes. The video must be watched to answer.
4. **Answer Certainty and Grounding:** Each question must have an unambiguous correct answer that is fully inferable from the structured annotations. Do not include speculative or uncertain phrasing in the answers. Avoid terms like "maybe," "appears to," or "possibly."
5. **No Internal Identifiers or Serial Labels:** Do not use labels such as "Person A," "Rope 1," or "Frame 3" in questions or answers. Instead, refer to entities using their described attributes (e.g., "the thick red-and-blue rope," "the hand with pale skin and slender fingers"), use the scene or the activity to ground the time instead of the specific frames or second numbers, so that a viewer can ground them visually. Also, the segment information is not provided to the participants, so do not mention the segment index or time in the question or answer.

**Output Format**
Provide the output as a JSON string, structured as a Python List and each entry is a Python Dictionary. Do not include a comma at the end of the last entry.
Example Output:

> [
>   {"Question": "<question-1>", "Answer": "<answer-1>"},
>   ...
>   {"Question": "<question-N>", "Answer": "<answer-N>"}
> ]

> [
>   {"Question": "<question-1>", "Answer": "<answer-1>", "Distractors": ["<distractor-1-1>", "<distractor-1-2>", "<distractor-1-3>"]},
>   ...
>   {"Question": "<question-N>", "Answer": "<answer-N>", "Distractors": ["<distractor-N-1>", "<distractor-N-2>", "<distractor-N-3>"]}
> ]

**Attention:**
1. Output must be pure JSON without any markdown code blocks or other formatting.
2. The output must be valid JSON without any comments. No ellipsis (...) or comments (//) about similar patterns.
3. If no relevant QA pairs can be generated, simply return the word 'None' in string format.

Figure 7: The prompt used for behavior-based data synthesis with `Gemini-1.5-pro`. The texts in the yellow block are used for open-ended QA synthesis, while the texts in blue blocks are used for Multiple-Choices Question synthesis. The `question_scope` is replaced with concrete descriptions in Figure 10 in the implementation.

# D Additional Experiments and Ablations

In this section, we complement the main results with additional experiments: (i) expanded baselines on egocentric tasks, (ii) new benchmarks beyond Ego4D, (iii) ablations on token compression and curriculum strategies, (iv) cost–benefit of token compression.

Table 7: Performance of expanded baselines on egocentric benchmarks. Numbers are the accuracy (%). The performance gains over Qwen2.5-VL are reported in parentheses.

| Method | EgoSchema | EgoPlan | QAEgo4D | EgoTaskVQA | OMEBench (beh.) | OMEBench (vis.) |
|---|---|---|---|---|---|---|
| Gemini-2.0-Flash | **71.1** | **51.5** | 62.4 | **59.2** | 60.2 | 62.1 |
| AlanVLM | 50.7 | 27.6 | 41.8 | 43.2 | 27.3 | 33.2 |
| Qwen2.5-VL | 65.0 | 45.2 | 59.4 | 53.7 | 55.3 | 49.4 |
| VideoChat-Flash | 64.3 | 40.8 | 62.4 | 54.8 | 57.1 | 54.5 |
| **OpenMMEgo (Qwen2.5-VL)** | 69.3 (+4.3) | 50.2 (+5.0) | **65.6** (+6.2) | 56.2 (+2.5) | **65.7** (+10.4) | **63.2** (+13.8) |

Table 8: Results on new benchmarks. Numbers are accuracy (%) except that EP-100 MIR is mAP (%).

| Method | HD-EPIC | OpenEQA | EP-100 MIR |
|---|---|---|---|
| EgoVLP | – | – | 26.0 |
| LaVila-L | – | – | 40.0 |
| Gemini-2.0-Flash | 38.8 | 61.5 | 49.3 |
| AlanVLM | 25.7 | 46.7 | 23.1 |
| Qwen2.5-VL | 34.2 | 55.7 | 42.1 |
| VideoChat-Flash | 39.4 | 58.1 | 45.3 |
| **OpenMMEgo (Qwen2.5-VL)** | **42.4** (+8.2) | **60.2** (+4.5) | **49.7** (+7.6) |

## D.1 Expanded Baselines and Benchmarks

**Additional Baselines.** We introduce two additional baselines to better situate OpenMMEgo among open and closed models. First, since the **Gemini-1.5-Pro** API was no longer available at evaluation time and its scale is much larger than our 7B backbone, we instead report results of **Gemini-2.0-Flash**. This model is closer in size to our backbone, and it is reasonable to assume that its training data and process are highly similar to Gemini-1.5-Pro, making it a suitable closed-source reference. Second, we include **AlanVLM** (Suglia et al., 2024), which is trained on synthetic dataset, as a supplementary baseline for using synthetic egocentric data. Results are shown in Table 7, where OpenMMEgo consistently improves upon all open baselines, narrowing the gap with Gemini despite being trained with fewer resources.

**Additional Benchmarks.** Beyond Ego4D-based suites, we further evaluate OpenMMEgo on several newly added benchmarks to examine its generalization ability. Specifically, we include results on the **HD-EPIC VQA** (Perrett et al., 2025) benchmark, which emphasizes fine-grained egocentric video question answering, and the open-ended **OpenEQA** (Majumdar et al., 2024), which tests broader video-language reasoning in unconstrained settings. In addition, we adapt the **EPIC-Kitchens-100** (Damen et al., 2022) multi-instance retrieval ($V \rightarrow T$) task into a multiple-choice format, enabling a fairer comparison with video-language models; performance is reported in terms of mAP. For this retrieval setting, we also report results from **EgoVLP** (Lin et al., 2022b) and **LaViLa-L** (Zhao et al., 2023) as strong references. Together, these benchmarks complement Ego4D tasks by providing both fine-grained and open-domain evaluation scenarios, thereby offering a more comprehensive assessment of egocentric video understanding. Table 8 shows that OpenMMEgo achieves consistent improvements over base LMMs, e.g. +8.2% on HD-EPIC and +7.6% mAP on EP-100, indicating robustness beyond Ego4D.

## D.2 Strategy Ablations

We conduct the additional ablation experiments on the hyperparameter $\alpha$, $k$,$r_l$, and $r_h$ to investigate their influences on OpenMMEgo. We also include the comparison with alternatives to our design in token compression and curriculum learning. Given the limited computation resources, the Open-MMEgo variants in this section are all trained with 1.6M pieces of data sampled from the overall OME10M.

**Online Dropout Ablation.** The hyperparameter $\alpha$ governs the proportion of units dropped out in each batch during online learning. This parameter is intended to encourage the model to focus on the more readily learnable aspects of the current stage during fine-tuning. As the results in the Table 9 indicate, performance suffers when $\alpha$ is either too small or too large. However, values around 0.3

Table 9: The results of additional ablation experiments on the hyperparameters $\alpha$, $k$, $r_l$, and $r_h$ of OpenMMEgo with OpenMMEgo-Qwen2.5-VL. We report the accuracy (%) of multiple-choice QAs on 5 egocentric video benchmarks, and the average accuracy (%) on 3 general video benchmarks (Video-MME, MVBench, and PerceptionTest) is reported under 'Gen. Ben.'.

| | EgoSchema | EgoPlan | QAEgo4D | EgoTaskVQA | OMEBench (beh.) | OMEBench (vis.) | Gen. Ben. |
|---|---|---|---|---|---|---|---|
| # Ablations on Online Dropout | | | | | | | |
| $\alpha = 0.05$ | 65.1 | 47.1 | 60.4 | 53.8 | 57.6 | 56.9 | 63.7 |
| $\alpha = 0.3$ | 66.5 | 47.4 | 62.6 | 53.9 | 62.1 | 58.0 | 68.5 |
| $\alpha = 0.5$ | 66.7 | 47.7 | 61.4 | 52.8 | 61.6 | 58.2 | 68.1 |
| $\alpha = 1.0$ | 64.2 | 45.8 | 58.6 | 50.6 | 51.5 | 53.2 | 62.2 |
| # Ablations on STM | | | | | | | |
| $k = 10$ | 65.1 | 45.7 | 61.4 | 52.8 | 58.6 | 55.3 | 66.3 |
| $k = 35$ | 66.5 | 47.4 | 62.6 | 53.9 | 62.1 | 58.0 | 68.5 |
| $k = 50$ | 64.7 | 46.1 | 60.4 | 52.6 | 55.7 | 57.3 | 67.8 |
| # Ablations on TTP | | | | | | | |
| $r_l = 0, r_h = 60$ | 59.3 | 45.1 | 58.4 | 47.8 | 55.3 | 54.1 | 62.3 |
| $r_l = 15, r_h = 75$ | 65.1 | 46.1 | 60.4 | 51.8 | 60.7 | 57.0 | 67.9 |
| $r_l = 35, r_h = 95$ | 66.5 | 47.4 | 62.6 | 53.9 | 62.1 | 58.0 | 68.5 |
| $r_l = 40, r_h = 100$ | 63.2 | 45.1 | 60.4 | 50.8 | 57.3 | 56.1 | 63.7 |

Table 10: Ablation on alternative compression and curriculum strategies. We compare OpenMMEgo with two alternatives: (i) VTM, a learnable spatiotemporal merging baseline for token compression, and (ii) a standard short→long curriculum. Numbers are accuracy (%) except that EP-100 MIR is mAP (%).

| Benchmark | OpenMMEgo | w. VTM | w. short→long |
|---|---|---|---|
| EgoSchema | **66.5** | 65.8 | 65.3 |
| EgoPlan | **47.7** | 46.3 | 45.9 |
| QAEgo4D | **62.6** | 61.8 | 62.4 |
| EgoTaskVQA | **53.9** | 52.1 | 53.7 |
| OMEBench (beh.) | **62.1** | 60.4 | 58.7 |
| OMEBench (vis.) | **58.0** | 55.7 | 56.2 |
| HD-EPIC | **36.1** | 34.2 | 35.4 |
| OpenEQA | **58.4** | 54.3 | 53.1 |
| EP-100 MIR (mAP) | **46.1** | 45.2 | 45.7 |

(such as 0.5) have a limited impact on performance. This suggests that online dropout can effectively improve performance as long as the dropout ratio remains within a reasonable range.

**STM Ablation.** The hyperparameter $k$ governs the compression ratio in spatial token compression. With experiments conducted using the upper limit of input frame count, the results in the Table 9 indicate that both excessively high and low compression ratios lead to performance degradation. This suggests that aggregating semantic entity information is effective, and a granularity of 35% is a suitable level for aggregating semantic entities.

**TTP Ablation.** The hyperparameters $r_l$ and $r_h$ are related to the compression ratio in temporal token compression. The experiments in Section 5.3 have already demonstrated the effectiveness of temporal compression. Here, we keep the token pruning ratio constant to investigate the impact of different compositions of high and low pruning similarity on the experiment. As shown in the Table 9, the model's performance is relatively sensitive to this parameter. A possible reason is that excessive pruning of tokens with high similarity will cause semantic loss, while excessive pruning of tokens with low similarity may lead to loss of temporal information.

**Strategy Alternatives.** We also test alternatives to our design, including **VTM** (Lee et al., 2024) (learnable spatiotemporal merging) for compression and a **short→long** curriculum. As shown in Table 10, both underperform our dual strategy: VTM drops entity-awareness and hurts OMEBench, while short→long yields only minor gains, confirming the effectiveness of our difficulty-aware curriculum.

# E  Additional Visualization on Visual Token Compression

For a better understanding of the visual token compression process in different scenarios, we here provide more visualization examples of the visual token compression process in OpenMMEgo-LLaVA-Video. As shown in Figure 12, the videos vary in the distance of the activity area and the degree of perspective change. Our method effectively compresses the visual tokens in the video while retaining important semantic information across these different scenarios.

# F  Limitation Discussion

While OpenMMEgo enhances large multimodal models' (LMMs) ability to understand egocentric videos by constructing a large-scale synthetic dataset, this data-driven process implicitly injects certain reasoning priors into the models. In the context of the recent surge in interest toward reasoning capabilities, a natural question arises: can these reasoning abilities be further improved through reinforcement learning (RL), particularly tailored for egocentric scenarios? At present, this direction remains unexplored in our experiments. Our observation is that the effectiveness of RL in eliciting reasoning hinges on the model's ability to produce basic reasoning-oriented outputs as a starting point. However, current LMMs still exhibit notable limitations in egocentric video reasoning, which may hinder RL-based enhancement. From this perspective, our dataset serves to equip LMMs with a foundational capacity for egocentric video understanding, potentially paving the way for future work that leverages reinforcement learning to induce more advanced, task-specific reasoning abilities.

**Input & Task**

You are provided with sequential frames extracted from an ego-centric video, with a consistent time interval of 1 seconds between adjacent frames. As an aid, some narrations are also provided to you. Some marks are used in the narrations: '#C' for the camera wearer, '#not sure' for the uncertain information, '#summary' for the summary of the narration sequence, '#O' for other action conductors. Your task is to analyze the scene and provide a response in JSON format which depicts the scene sets and the motion analysis of the scene in the ego-centric video.

**Guidelines:**

1. Object Identification and Description: Analyze all the frames to identify the objects in the scene and describe them, including their names, attributes (such as size, color, feature, condition, and any distinguishing characteristics), and positions. For each object that appears, provide:

    <object-entry>: { "Name": <string>, "Attributes": <string>, "Position": <string>}

2. Tracking Adjective Changes: For every pair of adjacent frames, document changes in each object's attributes and position. If an object is partially visible or temporarily disappears, infer and describe its motion, including the reasoning. Additionally, classify each object into one of the following categories:

    **a) Environmental Background**: Stationary elements in the scene whose apparent movement is due to camera motion.
    **b) Interactive Object**: Objects manipulated or used by the camera wearer, often moving closer or into view.
    **c) Moving Passerby**: Objects or people moving independently of the camera wearer.

For each entity, provide:

    <entity-change-entry>: { "Name": <string>, "Change": <string>, "Visibility": <string>, "Type": <string>}

3. Camera Movement Analysis: Infer the camera wearer's movement from changes in perspective across frames, considering shifts in various object types. Describe the trajectory, including direction, angle, and speed variations. Deduce possible body part movements (e.g., head, arms, legs) based on these perspective changes and available narration. For each adjacent frame pair, provide:

    <camera-movement-entry>: {"Perspective": <string>, "Camera Movement":<string>, "Body Part Movement": <string>}

4. Scene Summary: After processing all frames, synthesize a comprehensive scene summary. Include:

    **a)** A structured spatial description of the environment and object layout, with reference to logical groupings
    **b)** A summary of the camera wearer's motion across the scene, detailing trajectory and changes in orientation or speed.
    **c)** A summary of inferred body movements throughout the scene.

For each video clip, provide:

    <scene-summary-entry>:{"Scene": <string>,"Camera Movement": <string>, "Body Part Movement ":<string>}

5.Interaction Analysis: Describe all interactions between the camera wearer and objects in the environment. Include every phase of interaction (e.g., approaching the object, manipulating it, relocating it). Clearly describe: object characteristics, affordance, specific body parts and tools involved, purpose of interaction, detailed motion trajectory (including precise angles, distances, speed, path curvature, rotation, depth changes, and spatial shifts) For each interaction, provide:

    <interaction-entry>: {"Interaction": <string>, "Start Time": <int>,"End Time": <int>, "Description": <string>}

**Output Format**

A JSON formatted output:

```
{
    "Entities": [ <object-entry-1>, <object-entry-2>, ... ],
    "Adjective Changes": [
        "0,1": {
            "Entity": [<entity-change-entry-1>, <entity-change-entry-2>, ...],
            "Camera Movement": <camera-movement-entry>,
        }
        "1,2": {
            "Entity": [<entity-change-entry-1>, <entity-change-entry-2>, ...],
            "Camera Movement": <camera-movement-entry>,
        }
        ...
    ],
    "Scene Summary": <scene-summary-entry-1>,
    "Interactions": [<interaction-entry-1>, <interaction-entry-2>, ...],
}
```

**Key Requirements:**

1. Output must be pure JSON without any markdown code blocks or other formatting.
2. All information must be explicitly stated. Take each adjacent frame pair into consideration for motion and spatial analysis.
3. All the descriptions must be detailed and specific, avoiding generic terms.
4. The output must be valid JSON without any comments. Every object's motion must be fully described for each time interval, without any omissions or shortcuts. No ellipsis (...) or comments (//) about similar patterns.

Figure 8: The prompt used for vision annotation with `Gemini-1.5-pro`.

**Input & Task**

You are tasked with generating high-quality `Question-Answer (QA) pairs` `Multiple-Choice Questions (MCQs)` to evaluate participants' understanding of egocentric videos. Instead of raw video input, you are provided with structured descriptions of each video segment, spanning durations from 10 seconds to several minutes. Each segment is divided into 10-second intervals and annotated with:

- A detailed list of **Entities**, including their **Names**, **Attributes** (e.g., color, material, size, status), and **Positions** (relative to the egocentric view).
- Per-second **Adjective Changes** for each entity: tracking movement, interaction, rotation, occlusion, and visibility changes.
- Detailed **Camera Movement** and **Body Part Movement** per second: covering perspective shifts, self-motion, and viewpoint adjustments.
- A high-level **Scene Summary** and **Interaction Summary**: describing ongoing actions, spatial layout, object interactivity, and bodily involvement.

As an assistance, some concise human annotations of the atomic actions are also provided to you. Some marks are used in the narrations: '#C' for the camera wearer, '#not sure' for the uncertain information, '#summary' for the summary of the narration sequence, '#O' for other action conductors.

Your task is to generate question-answer pairs that probe the participant's comprehension of visual, temporal, spatial, causal, motor, and interactional aspects of egocentric video, based mainly on the annotated visual information and slightly on the human annotations.

> **The concrete visual attributes of the objects and the spatial-temporal relationships between them should be integrated in the answers.**

> **Each question must have one correct answer and three plausible distractors**. The distractors should be **confusing but incorrect**, grounded in visual elements that appear in the video but do not satisfy the question, or artificial elements that could reasonably be inferred but are wrong.

**Guidelines:**

**1. Question Scope**: You should generate questions about **{question_scope}**. You should generate appropriate questions in these angles or their composition.

**2. Question-Answer Quality:**

> The generated questions should vary in complexity and scope. Across the generated QA pairs, ensure coverage of diverse question angles within the assigned dimension. Avoid producing repetitive patterns or overly similar question formulations. Some questions should target fine-grained visual details and result in short, factual answers. Others should require reasoning across multiple temporal segments or entities, resulting in longer, integrative answers. Both short-form and long-form questions must be present in each generation batch. For the diversity of questions, only part of the information should be involved in each QA pair. Though the answers may be long, the questions should be short and concise to hit the nail on the head.

> The generated questions should vary in complexity and scope. Across the generated MCQs, ensure coverage of diverse question angles within the assigned dimension. Avoid producing repetitive patterns or overly similar question formulations. Some questions should target fine-grained visual details and others should require reasoning across multiple temporal segments or entities. For the distractors:
> a) Distractors must be plausible in the scene or event.
> b) They should differ subtly from the correct answer in **visual facts**, e.g., a wrong object, wrong timing, wrong attribute, or slightly incorrect reasoning.
> c) Avoid obviously false or out-of-scope distractors.

**3. No Answer Leakage:** The answer must not be directly inferable from the question text alone. Avoid phrasing that embeds hints, reuses descriptive terms from annotations, or overly specifies entities or outcomes. The video must be watched to answer.

**4. Answer Certainty and Grounding:** Each question must have an unambiguous correct answer that is fully inferable from the structured annotations. Do not include speculative or uncertain phrasing in the answers. Avoid terms like "maybe," "appears to," or "possibly."

**5. No Internal Identifiers or Serial Labels:** Do not use labels such as "Person A," "Rope 1," or "Frame 3" in questions or answers. Instead, refer to entities using their described attributes (e.g., "the thick red-and-blue rope," "the hand with pale skin and slender fingers"), use the scene or the activity to ground the time instead of the specific frames or second numbers, so that a viewer can ground them visually. Also, the segment information is not provided to the participants, so do not mention the segment index or time in the question or answer.

**Output Format**

Provide the output as a JSON string, structured as a Python List and each entry is a Python Dictionary. Do not include a comma at the end of the last entry.

Example Output:

```
[
  {"Question": "<question-1>", "Answer": "<answer-1>"},
  ...
  {"Question": "<question-N>", "Answer": "<answer-N>"}
]
```

```
[
  {"Question": "<question-1>", "Answer": "<answer-1>", "Distractors":
    ["<distractor-1-1>", "<distractor-1-2>", "<distractor-1-3>"]},
  ...
  {"Question": "<question-N>", "Answer": "<answer-N>", "Distractors":
    ["<distractor-N-1>", "<distractor-N-2>", "<distractor-N-3>"]}
]
```

**Attention:**
1. Output must be pure JSON without any markdown code blocks or other formatting.
2. The output must be valid JSON without any comments. No ellipsis (...) or comments (//) about similar patterns.
3. If no relevant QA pairs can be generated, simply return the word 'None' in string format.

Figure 9: The prompt used for vision-centric data synthesis with `Gemini-1.5-pro`. The texts in the yellow block are used for open-ended QA synthesis, while the texts in blue blocks are used for Multiple-Choices Question synthesis. The `question_scope` is replaced with concrete descriptions in Figure 10 in the implementation.

**Atomic Actions,** including:
- Action recognition: Identifying specific movements or manipulations mentioned in the narration.
- Interactive object grounding: Determining which objects are involved in a specific action.
- Temporal reasoning: Understanding action durations, sequences, or changes over time.
- Action counting: Inferring how many times an action occurred, or comparing frequencies.

**Chunked Behaviors**, where atomic actions are grouped into higher-level descriptions. These questions should:
- Aggregate sequences of atomic actions into meaningful behavioral units.
- Ask for interpretations of overall behavior (e.g., preparing tea, cleaning up).
- Compare or contrast different behavior phases or repetitions.

**Goal-Step reasoning**, including:
- Goal analysis: Inferring the likely objective based on behavior.
- Next-step prediction: Predicting what should logically happen next to complete the task.
- Precondition identification: Determining what must have already occurred for the next step to proceed.

**Open Form**. Generate unique and creative questions specific to the given egocentric video segment. These questions should go beyond generic templates and instead target the most distinctive, non-obvious, or surprising aspects observed in the structured annotations. context-sensitive questions that are specific to the unique content of the narrated video:
- Identifying an unexpected behavior or anomaly occurring in the scene.
- Spotting subtle or unusual interaction sequences not typical in everyday contexts.
- Reasoning about cause-effect relationships that are unique to this video (e.g., why a certain reaction happened after a minor motion).
- Test story-like comprehension or complex causal chains.

**Spatial-Temporal Reasoning.** These questions aim to assess understanding of spatial relationships of objects and their evolution over time. They should probe the participant's ability to track object positions, relative displacements, and visual transitions, considering both local and global temporal contexts. Allowed question angles are as follow:
• Tracking the spatial movement of a single entity over time
• Comparing the relative positions of multiple entities across different time points
• Detecting consistent motion trends, such as convergence, divergence, or rotational drift
• Inferring whether observed movements are caused by camera motion or object motion
• Identifying spatial transformations or layout shifts of objects and surfaces
• Reasoning about positional symmetry or alignment between objects at different moments
• Any visual reasoning that involves the coupling of temporal change and spatial layout

**Self-Motion Understanding.** Questions should focus on interpreting self-induced motion patterns, including both intentional actions and subtle balance adjustments. Allowed question angles include:
• Detecting changes in body pose (e.g., sitting, standing, leaning, turning)
• Inferring head or torso rotation and the resulting shift in visual perspective
• Analyzing hand movement trajectories in relation to torso/camera motion
• Identifying transitions between static and dynamic bodily states
• Detecting compensatory movements for balance (e.g., shifting weight before or after an action)
• Evaluating how visual stability is maintained or disturbed across time
• Any reasoning involving bodily posture, physical coordination, or proprioception from first-person view

**Interactions**. These interactions may be tool-based, hand-driven, or involve multi-object manipulation. Allowed question angles include:
• Describing fine-grained manipulation of a specific object (e.g., gripping, rotating, pulling)
• Analyzing two-handed coordination on a single object or across multiple items
• Inferring mechanical or functional changes caused by interaction (e.g., rope tightening, lid opening)
• Identifying when and how objects are combined, detached, or used in sequence
• Detecting interaction phases (initiation, mid-action, completion) and associated feedback
• Recognizing when tool affordances are being utilized or repurposed
• Any inference involving dynamic physical contact or control over objects

**Vision-Inferred Behavior**. High-level reasoning about intentions, planning, affordances, and goals. These should be grounded in visible cues and contextual transitions, avoiding speculative or purely textual logic. Allowed question angles include:
• Inferring the goal or purpose of an action based on current visual and physical context.
• Determining whether an object's properties (e.g., position, material, accessibility) afford a specific action.
• Assessing the sequencing of actions to evaluate goal-directed planning or preparatory behavior.
• Reasoning about behavioral changes in response to visual scene dynamics (e.g., a new object entering view).
• Judging whether a particular behavior has achieved its intended result, using visual evidence.
• Predicting plausible next actions based on ongoing activity and object affordances.
• Connecting current actions to previously observed steps in a longer behavioral chain.
• Any reasoning that connects perceptual cues with implied agent intention, planning, or task logic.

Figure 10: The detailed description of question scopes used in behavior-based data synthesis (top) and vision-centric data synthesis (bottom).

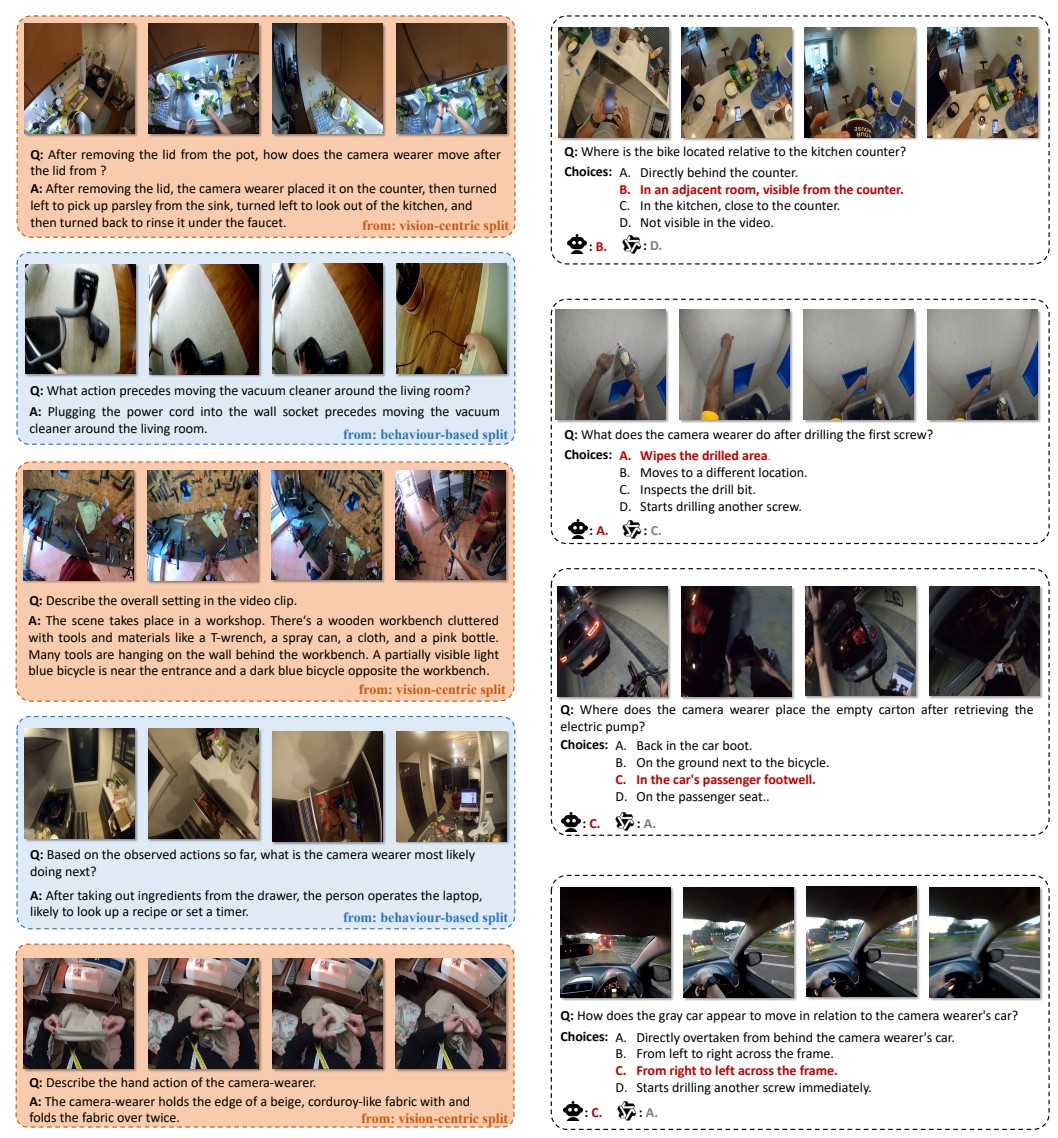

Figure 11: Examples from OME10M (Left) and OMEBench (Right). The QA pairs from dual splits of OME10Mare on the left. Examples from OMEBench and the corresponding answers from OpenMMEgo-Qwen2.5-VL and Qwen2.5-VL are provided on the right.

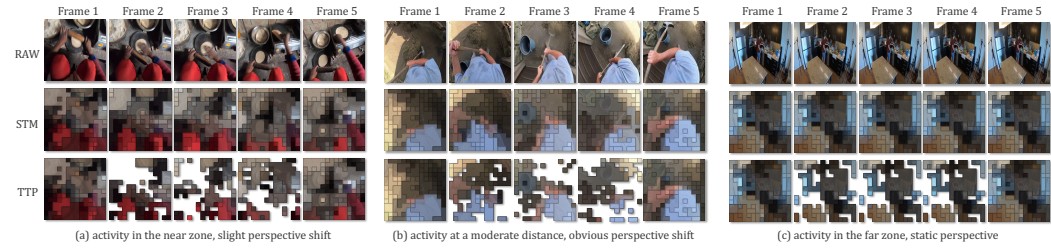

Figure 12: Visualization of the visual token compression process of OpenMMEgo-LLaVA-Video. RAW refers to the original frames from videos. STM and TTP represent the visual tokens after Spatial-redundant Token Merging and Temporal-irrelevant Token Pruning, respectively.

