# OpenReview forum: "OpenMMEgo: Enhancing Egocentric Understanding for LMMs with Open Weights and Data"
_NeurIPS.cc/2025/Conference — NeurIPS 2025 poster_

### Official Review · Reviewer_3kaY · 2025-06-26

**Clarity:** 3
**Significance:** 3
**Originality:** 2
**Rating:** 4
**Confidence:** 5

**Summary:**

The paper introduces OpenMMEgo, a recipe for improving large multimodal models (LMMs) on first-person (egocentric) video. The authors curate OME10M and OMEBench for model training and evaluation. The authors propose a Dual Semantic-aware Token Compression module that compresses long videos by (i) Spatial-redundant Token Merging (STM) and (ii) Temporal-irrelevant Token Pruning (TTP), aiming to keep informative tokens while discarding redundancy. Moreover, a Dual Curriculum Learning Strategy is introduced to combine an offline difficulty curriculum and an online loss-based dropout to stabilize learning on diverse data.

**Questions:**

Please refer to the weakness section.

**Ethical Concerns:**

["NO or VERY MINOR ethics concerns only"]

**Final Justification:**

I have read the rebuttal and would like to keep my original rating.

**Limitations:**

Yes.

**Paper Formatting Concerns:**

No.

**Quality:**

3

**Strengths And Weaknesses:**

### Strengths:

1. Comprehensive egocentric dataset & benchmark

OME10M and OMEBench fill a gap of the study of egocentric video understanding: public, large-scale resources for first-person video QA, with multi-level visual supervision.

2. Thorough empirical evaluation

Results cover five external egocentric benchmarks plus three general-video suites, and ablations dissect data components, curricula, and compression variants.

3. Marginal degradation on generic tasks

The proposed egocentric tuning slightly harms general video understanding.

### Weaknesses:

1. Incremental technical novelty

STM builds on existing token-merging work, and TTP mirrors SlowFast-style sampling; both ideas are adaptations rather than fundamentally new. Similarly, offline/online curricula echo established schedules. The conceptual advance may be viewed as a limited aggregation of prior techniques.

2. Unclear computation cost of STM and TTP
Although STM and TTP can remove redundant tokens, their computational cost is underexplored. It remains unclear how the additional operations introduced by STM and TTP affect inference speed.

3. Limited analysis of failure modes

The paper provides no qualitative error analysis or robustness checks.

4. Limited comparison between different egocentric datasets

The reviewer appreciates the efforts of curating the OME10M dataset. It is recommended to compare OME10M with existing general video or egocentric video datasets to emphasize the contribution or specialty of the OME10M dataset.

---

> ### Author Rebuttal · Authors · 2025-07-31
>
> Thank the reviewer for the thoughtful and balanced assessment of our work. We are grateful for the recognition of OME10M and OMEBench as valuable large-scale resources for egocentric video QA, as well as for the thorough empirical evaluation across multiple benchmarks and ablation studies. Below, we address each of these points in detail.
>
>  **Technical novelty**: Indeed, some of the ideas in strategy we employed have appeared in previous work, but our design is tailored to the OME10M dataset. Determining which training strategies are more effective for learning on multi-level visual supervision data is a topic worth exploring. On one hand, our outperformance over the baselines to some extent demonstrates the effectiveness of our designs (for instance, VideoChat-Flash also employs various video token compression strategies). Additionally, we introduced further experiments, using VTM as an alternative for video token merging and adopting "from short videos to long videos" as a straightforward curriculum learning baseline. The experimental results on the 1.6M dataset are shown in the table below. These results highlight the value of investigating appropriate training strategies for LMMs and datasets.
>
> |                        | EgoSchema | EgoPlan | QAEgo4D | EgoTaskVQA | OMEBench (beh.) | OMEBench (vis.) | HD-Epic | OpenEQA | EP-100 MIR(V →T) |
> | ---------------------- | --------- | ------- | ------- | ---------- | --------------- | --------------- | ------- | ------- | ---------------- |
> | OpenMMEgo              | 66.5      | 47.7    | 62.6    | 53.9       | 62.1            | 58.0            | 36.1    | 58.4    | 46.1             |
> | OpenMMEgo w. VTM       | 65.8      | 46.3    | 61.8    | 52.1       | 60.4            | 55.7            | 34.2    | 54.3    | 45.2             |
> | OpenMMEgo w. shot→long | 65.3      | 45.9    | 62.4    | 53.7       | 58.7            | 56.2            | 35.4    | 53.1    | 45.7             |
>
> **Computation Cost of STM and TTP**
> Indeed, during the process of token compression, additional operations introduce new time costs. To better illustrate the extra time cost of token compression, we use the inference time of OpenMMEgo without STM+TTP as the unit, comparing the costs of different operations with the maximum number of frames under the same memory.
>
>
> |                | w/o STM+TTP | w/o STM | w/o TTP | OpenMMEgo |
> | -------------- | ----------- | ------- | ------- | --------- |
> | frames         | 64          | 106     | 91      | 192       |
> | inference cost | 1.0         | 1.31    | 1.19    | 1.47      |
>
> Our token compression strategy enables processing **two times additional frames** with only a 47 % increase in inference time.
>
> **Analysis of Failure Modes**: We thank the reviewer for the suggestion. Due to the limitations of not being able to include images in the response, we shall address this point in the next version of the paper by incorporating erroneous examples and the difference in results from multiple inferences.

---

### Official Review · Reviewer_FPzo · 2025-06-29

**Clarity:** 3
**Significance:** 3
**Originality:** 4
**Rating:** 5
**Confidence:** 2

**Summary:**

This paper introduces OpenMMEgo, a family of egocentric large multimodal models (EgoLMMs). It also presents OME10M, a large-scale fine-tuning egocentric dataset with 8.2M egocentric video QA pairs from Ego4D, and OMEBench, a benchmark for egocentric understanding. The authors propose Dual Semantic-aware Token Compression, an egocentric representation compression method comprising two modules: Spatial-redundant Token Merging (STM) and Temporal-irrelevant Token Pruning (TTP). Additionally, the work introduces two enhancements to Dual Curriculum Learning Strategy: Offline Data Curriculum and Online Data Dropout. Experiments demonstrate that OpenMMEgo elevates Qwen2.5-VL to state-of-the-art performance in egocentric understanding.

**Questions:**

OpenMMEgo achieves similar performance to other methods on video understanding tasks, but outperforms other methods on egocentric video understanding tasks. The paper discusses some theories behind it, such as egocentric vision requires rich low-level visual cues. Are there any analysis and experiments that helps explain this?

**Ethical Concerns:**

["NO or VERY MINOR ethics concerns only"]

**Final Justification:**

The rebuttal addressed my questions and I will maintain my original rating.

**Limitations:**

Yes

**Quality:**

4

**Strengths And Weaknesses:**

Strengths:
1. This paper offers good technical novelty by introducing improvements in model architecture and training strategy.

It proposes Spatial-redundant Token Merging (STM) and Temporal-irrelevant Token Pruning (TTP), which together reduce redundancy and noise in video representation learning.

For training strategy, it introduces Offline Data Curriculum and Online Data Dropout, quantifying sample difficulty using loss and dynamically filtering challenging in-batch samples during training.

2. OpenMMEgo achieved comparable performance with existing methods in general video understanding tasks, and achieves state-of-the-art in egocentric video understanding tasks.

3. The paper includes ablation studies that validate the effectiveness of the proposed methods.

4. All components of this work will be open sourced, promoting transparency and collaboration in this field.

Weakness:
No major weakness was identified.

Minor weaknesses:
Line 35: Missing comma after OpenMMEgo?

Some designs are under-explained in the main paper, such as the curation of OME10M, likely due to page limitations.

---

> ### Author Rebuttal · Authors · 2025-07-31
>
> We thank the reviewer for the thoughtful and constructive feedback. We reply to the concerns below and will incorporate the mentioned revisions in the next version of our manuscript.
>
> Line 35: Indeed, a comma is missing at this position and will be inserted in the revised manuscript.
>
> **Data Curation Details**: The main text outlines only the core design concepts due to space constraints. A comprehensive description of the OME10M generation pipeline appears in Appendix B, covering the end-to-end workflow, all categories of synthetic QA pairs, and the exact prompt templates employed for behavior-based and vision-centric question creation.
>
> **The Role of Visual Cues**: Intuitively, for general understanding, third-person training data inherently contains descriptions of visual details, which are primarily lacking in egocentric video data. Therefore, introducing annotations of visual details from egocentric video makes the data distribution more uniform and thus easier to learn. On the other hand, the inference of events and actions in egocentric video often relies on these visual details. When egocentric data lacks visual details, general understanding and egocentric video understanding become significantly different tasks and are hard to foster each other. Experimentally, we removed the visual details data in ablation studies, which resulted in a comprehensive performance decline.
>
> These clarifications will be incorporated into the revised version to enhance reproducibility and to shed light on why visual-detail supervision is essential for egocentric video understanding.
>
> We hope these clarifications resolve the reviewer's concerns. Please let us know if any further questions.

---

> > ### Comment · Reviewer_FPzo · 2025-08-05
> >
> > Thanks for the clarifications and agreeing to revise the comma.

---

> > > ### Author Response · Authors · 2025-08-05
> > >
> > > We thank the reviewer for the further response and for taking the time to review our work. If there are any further concerns or questions, please feel free to let us know.

---

### Official Review · Reviewer_pEBA · 2025-07-01

**Clarity:** 3
**Significance:** 4
**Originality:** 3
**Rating:** 5
**Confidence:** 4

**Summary:**

In the field of egocentric video understanding, this paper presents OpenMMEgo, a large multimodal model that addresses challenges across three dimensions: data curation, model architecture, and training strategy. To mitigate frequent viewpoint shifts, the study introduces two key strategies: semantic-aware visual token compression and a curriculum learning framework. Additionally, the authors curate a new dataset and benchmark, named OME10M and OMEBench, respectively. Experimental results show that OpenMMEgo achieves significant performance gains on multiple egocentric benchmarks.

**Questions:**

1. Previous egocentric LMM studies often rely on Ego4D-based data augmentation. The paper could benefit from an in-depth comparison of different data construction strategies and their impact on model performance.

2. While the Spatial-Temporal Merging (STM) and Temporal Token Pruning (TTP) methods aim to reduce redundancy, the paper does not explicitly report computational costs (e.g., inference time, memory usage) compared to baseline models.

**Ethical Concerns:**

["NO or VERY MINOR ethics concerns only"]

**Final Justification:**

After reading the rebuttal and other reviews, there are no remaining concerns. I raised my rating accordingly.

**Limitations:**

Yes

**Quality:**

4

**Strengths And Weaknesses:**

Strengths:

1. OpenMMEgo proposes a multi-faceted approach, including: (i) Hierarchical data curation (both high-level and low-level) to enhance semantic richness. (ii) A model architecture designed to reduce computational overhead. (iii) Training strategies that optimize both stability and effectiveness.

2. The model demonstrates significant improvements on standard egocentric understanding benchmarks and shows comparable performance on general video understanding tasks, highlighting its generalizability.

3. Extensive ablation studies systematically validate the contribution of each core component, providing clear insights into the model’s design.

Weaknesses:

1. The writing in Section 3 (OME10M curation) lacks clarity, particularly regarding: (i) The specific methods used to construct behavior-based QAs and vision-centric QAs. (ii) The relationship between visual details annotations and the two QA types, making it difficult to replicate the curation process.

2. The paper omits recent advancements in egocentric research, including: (i) Benchmarks: EgoVQA, EgoThink, VidEgoThink, EgoPlan-2; (ii) LMMs: EgoVLP, EgoVLPv2, AlanaVLM.

3. A more comprehensive comparison with existing egocentric LMMs would strengthen the model’s performance validation.

---

> ### Author Rebuttal · Authors · 2025-07-31
>
> We thank the reviewer for the thoughtful and constructive feedback. We reply to the concerns below and will incorporate the mentioned revisions in the next version of our manuscript.
>
> **Data Curation Details**: Due to the limited space in the main text, we have only outlined the core design concepts. In Appendix B, we provide a detailed description of our data generation process, including the workflow, the specific types of data generated, and the prompts used. The visual details annotations are exclusively utilized in the visual details split of OME10M, and questions and answers are generated based on these annotations.
>
> **Additional Baselines & Benchmarks**
> We thank the reviewer for pointing out potential baselines and suggesting the non-Ego4d benchmark, which will indeed help us better present the performance of OpenMMEgo.
> - **AlanVLM**: We have included AlanVLM as an extra baseline, serving as a supplementary baseline for synthetic data.
> - **Video Language Pretrained Models**: We convert the multi-instance retrieval (V → T) task in Epic Kitchens 100 into a multiple-choice task to enable a comparison with  Video Language Pretrained Models (EgoVLP and LaViLa-L).
> - **New benchmarks:** We have added results from the HD-EPIC VQA benchmark and the open-ended OpenEQA.   Due to time constraints, EgoVQA and VidEgoThink have not yet completed testing, but they are in our plan and will be supplemented once finished.
>
> |                         | EgoSchema   | EgoPlan     | QAEgo4D         | EgoTaskVQA  | OMEBench (beh.)  | OMEBench (vis.)  | HD-Epic        | OpenEQA    | EP-100 MIR(V →T) |
> | ----------------------- | ----------- | ----------- | --------------- | ----------- | ---------------- | ---------------- | -------------- | ---------- | ---------------- |
> | EgoVLP                  | -           | -           | -               | -           | -                | -                | -              | -          | 26.0             |
> | LaViLa-L                | -           | -           | -               | -           | -                | -                | -              | -          | 40.0             ||
> | AlanVLM                 | 50.7        | 27.6        | 41.8            | 43.2        | 27.3             | 33.2             | 25.7           | 46.7       | 23.1             |
> | Qwen2.5-VL              | 65.0        | 45.2        | 59.4            | 53.7        | 55.3             | 49.4             | 34.2           | 55.7       | 42.1             |
> | VideoChat-Flash         | 64.3        | 40.8     | 62.4            | 54.8        | 57.1             | 54.5             | 39.4           | 58.1       | 45.3             |
> | OpenMMEgo（Qwen2.5-VL） | **69.3** (+4.3) | **50.2** (+5.0) | **65.6** (+6.2) | **56.2** (+2.5) | **65.7** (+10.4) | **63.2** (+13.8) | **42.4**(+8.2) | **60.2**(+4.5) | **49.7** (+7.6)  |
>
> OpenMMEgo consistently outperforms all other baselines, further demonstrating the effectiveness of our method.
>
> **Cost of STM and TTP**
> Indeed, during the process of token compression, additional operations introduce new costs. To better illustrate the extra time cost of token compression, we use the inference time of OpenMMEgo without STM+TTP as the unit, comparing the costs of different operations with the maximum number of frames under the same memory.
>
> |                | w/o STM+TTP | w/o STM | w/o TTP | OpenMMEgo |
> | -------------- | ----------- | ------- | ------- | --------- |
> | frames         | 64          | 106     | 91      | 192       |
> | inference cost | 1.0         | 1.31    | 1.19    | 1.47      |
>
> Our token compression strategy enables processing **two times additional frames** with only a 47 % increase in inference time.
>
>
> We hope these clarifications resolve the reviewer's concerns. Please let us know if any further questions.

---

### Official Review · Reviewer_iYek · 2025-07-02

**Clarity:** 2
**Significance:** 3
**Originality:** 3
**Rating:** 4
**Confidence:** 4

**Summary:**

This paper presents OpenMMEgo, an egocentric-focused VLM that is trained with novel data, model architecture and training strategies. In terms of data, the paper introduces OME10M, a large-scale (8.2M) synthetically-generated video QA dataset (for training) based on the Ego4D videos, and a corresponding evaluation split called OMEBench. On the modeling side, the paper performs two levels of semantic token compression to reduce the number of video tokens used in standard LLaVA-like architectures. Spatial-redundant Token Merging merges tokens within an image based on visual similarity-based connected components. Temporal-irrelevant Token Pruning follows a slow-fast strategy where slow frames retain all tokens and fast frames remove tokens that are too similar (i.e., redundant) or too different (i.e., irrelevant) to the previous slow frame. Finally, during training, the paper proposes a dual curriculum learning strategy that divides training data into different difficulty levels (offline) and trains on them one after the other. When training on a particular difficulty level, a hard-negative rejection strategy is used to downweight the probability of sampling hard samples. Experiments performed across different egocentric benchmarks demonstrate the benefits of the proposed method over several open-source baselines.

**Questions:**

I liked the paper in general because the ideas are interesting and the results are good. However, I have serious concerns that I would like the authors to address:
* Why is there a lack of comparison with related work and baselines suggested in the weakness section? These are basic and necessary to understand the contributions of the paper.
* Can we have more results on non-Ego4D video benchmarks?
* Could the authors clarify the missing details in the paper?

**Ethical Concerns:**

["NO or VERY MINOR ethics concerns only"]

**Final Justification:**

The authors addressed several of my concerns, but not some of my core concerns on token compression, synthetic annotations and baseline comparisons (summarized below). I've raised my rating to borderline accept (but not accept) because of these reasons.

# Remaining concerns

* **Token compression:** This is one of the core contributions (L56 - 62). The paper lacks a discussion of what prior token compression methods do and how this paper innovates over it. The authors have included one sample method in the rebuttal response, but it is unclear how this was selected and why other methods are not included. This needs a thorough study.

* **Synthetic annotations:** *LlavaVideo dataset* is used as a baseline, but the *approach for generating the LlavaVideo* dataset is not compared against. I would like to emphasize this difference. Simply using prior datasets in comparison is not what I'm suggesting. My key concern is that the data curation approach proposed here is not vetted against prior data curation approaches in an apples-to-apples fashion (same videos, same VLMs for annotations, but different curation process).

* **Baseline comparison is not apples-to-apples:** For context on Gemini 2.0 Flash vs. Gemini 1.5 Pro: Gemini 1.5 Pro is quite a bit better on video benchmarks as shown in Table 6 here: https://storage.googleapis.com/deepmind-media/gemini/gemini_v2_5_report.pdf. So the numbers reported here are likely worse than what 1.5 Pro would have scored.

**Limitations:**

Yes

**Quality:**

3

**Strengths And Weaknesses:**

# Strengths
* The paper proposes an interesting token merging approach that reduces number of video tokens spatially and temporally with benefits for overall performance.
* OME10M is curated sensibly and looks promising for the community to make progress on egocentric video understanding with VLMs
* The authors have promised to open-source the code and data (these are not included in the initial submission)
* Experiment design covers some essential egocentric video baselines and benchmarks (Table 1), and critical ablation studies indicating that each component of the model and data helps to varying degrees (Table 3, Figure 3) The results in Table 1 are promising since OpenMMEgo goes beyond MMEgo and provides large gains on the base VLM used for finetuning.

# Weaknesses
**Related work section is incomplete:** There are several related works on token merging and synthetic video QA generation. These have not been discussed sufficiently in the paper, and the benefits of the proposed method over these have not been explained nor studied.
* Token merging
	* ToME: https://arxiv.org/pdf/2210.09461
	* Token Merger: https://pubmed.ncbi.nlm.nih.gov/37440394/
	* LTMP: https://openreview.net/pdf?id=WYKTCKpImz
	* DTEM: https://arxiv.org/pdf/2412.10569
	* ToSA: https://arxiv.org/abs/2506.20066
	* STTS: https://www.ecva.net/papers/eccv_2022/papers_ECCV/papers/136950068.pdf
	* VTM: https://arxiv.org/pdf/2410.23782
	* InTI: https://arxiv.org/html/2408.06840v
* Synthetic annotations for egocentric videos
	* LaViLa: https://openaccess.thecvf.com/content/CVPR2023/papers/Zhao_Learning_Video_Representations_From_Large_Language_Models_CVPR_2023_paper.pdf
	* LLava video: https://llava-vl.github.io/blog/2024-09-30-llava-video/
	* EVUD: https://huggingface.co/datasets/AlanaAI/EVUD
	* Eagle 2.5: https://arxiv.org/pdf/2504.15271

**Missing baselines:** The experiment section neglects benchmarking alternate strategies for token merging, curriculum learning, and synthetic data annotation.
* **Token merging:** There are several works with varying motivations for reducing video tokens (listed above). The proposed STM and TTP strategies need to be benchmarked against these to verify that it is indeed useful.
* **Synthetic annotations:** The simplest synthetic annotations to generate are to ask a VLM like Gemini 1.5 Pro to generate questions given a video. There are more sophisticated strategies like Llava video and Eagle 2.5 that perform hierarchical captioning + QA generation, and LaViLa that generates dense narrations by training a narrator (which can then be used for QA generation). The proposed synthetic pipeline needs to be compared against these alternatives to show its benefits.
* **Curriculum learning:** A common strategy for training video models is to train on images first, short videos and then long videos (e.g., see https://arxiv.org/pdf/2501.00574v1 and https://www.arxiv.org/pdf/2408.10188). The proposed strategy based on sample difficulty is interesting, but simpler alternatives have not been compared against. Similarly, hard-negative filtering (L234) is interesting, but it is not clear how this compares to more traditional strategies like hard negative mining (or easy positive rejection) --- see page 2 here: https://lear.inrialpes.fr/people/triggs/pubs/Dalal-cvpr05.pdf.
* Gemini 1.5 Pro is used for generating annotations, but this model is not used as a baseline. Does the proposed approach go beyond Gemini 1.5 Pro? Or is it limited by the performance of the model (in which case, this would basically be a distillation from a closed-source model to the proposed model)?

**Benchmarks primarily focused on Ego4D:** The benchmarks in Section 5.2 are primarily focused on Ego4D. EgoPlan also includes a good part of Ego4D, and EgoTaskVQA is the only benchmark that goes outside of Ego4D. The results in Table 1 indicate that non-Ego4D benchmarks have the smallest gains (likely because the videos are out of domain from the training set). I would encourage the authors to include results on more non-Ego4D benchmarks to get a better sense of the broad applicability of the model. Here is a list (not comprehensive):
* EPIC Kitchens
* OpenEQA  (https://open-eqa.github.io/)
* EgoLifeQA (https://arxiv.org/abs/2503.03803)
* HD-EPIC (https://hd-epic.github.io/index#starthere)

**Missing details / clarifications needed:**
* **L171 - 172:** Is OMEBench human verified?
* **Section 4.1:** How expensive are the token merging operations? Can they be performed efficiently? How much time and compute does it take relative to the rest of the model operations?
* **L203:** How are tokens merged within a component?
* **L215:** "excessive semantic similarity" --- isn't this necessary to understand temporal movements of the object? Will removing redundant tokens like these cause a model to not understand basic flow of time?
* **L245 - 247:** Why not use the base model that is being finetuned for the difficulty estimation?
* **Figure 3:** The charts are a misleading here.
	* The left chart shows somewhat linear scaling of results, but the X axis is not linear. I would recommend a scatter plot with proper X scaling.
	* The right chart is very confusing because the scaling is different for each benchmark. For example, on EgoPlan, OpenMMEgo w/o STM performs exactly the same as EgoPlan, but a big margin is shown in the chart. I recommend having uniform scaling across all benchmarks to have a better understanding of the results.
* **Figure 5:** What is EgoCatch?

# Post-rebuttal update
Majority of my concerns were addressed, so I'm raising my rating to border-line accept. However, some core concerns on token compression, synthetic annotations and baseline comparisons were not sufficiently addressed. I'm not inclined to raise it further without these being addressed sufficiently.

---

> ### Author Rebuttal · Authors · 2025-07-31
>
> We thank the reviewer for the thoughtful and constructive feedback. We reply to the concerns below and will incorporate the mentioned revisions in the next version of our manuscript.
>
> **Potential Related Works**
> 1. **Token Compression techniques:** The reviewer listed many token merging methods developed for efficient visual representation learning, including ToMe, which we have cited in Line 194,  and ToSA, a paper posted after the NIPS submission Deadline. Our focus is on adapting token‐compression techniques (merging and pruning) in video understanding LMMs. Accordingly, we cited the classic origins (ToME for merging, LTP for pruning) in single-image and language-model contexts, and we discussed their extensions in video LMMs (e.g., VideoChat-Flash, LongUV, VideoLLaMA 3, etc.). VideoChat-Flash is also used as our baseline.
> 2. **Synthetic Annotations:** LLaVA-Video was indeed cited and is one of our baselines, based on which our SFT variant yields substantial gains. LaViLa focuses on visual-representation pretraining and thus differs in scope from LMMs. EVUD, a four-page unpublished manuscript, was inadvertently omitted, and Eagle 2.5 (a concurrent work) also went unnoticed.
>
> We appreciate the reviewer’s suggestions and will include discussions of all these works in the next revision as an additional Related Works section.
>
> ---
> **Additional Baselines & Benchmarks**
> We thank the reviewer for pointing out potential baselines and suggesting the non-Ego4d benchmark, which will indeed help us better present the performance of OpenMMEgo.
> - **Gemini Model Comparison**: Considering the scale difference with Gemini-1.5-Pro and the fact that the Gemini-1.5-Pro API is no longer available, we report the performance of the Gemini-2.0-Flash API on our egocentric benchmark as a comparison. The model size of Gemini-2.0-Flash is closer to our 7B model, and it is reasonable to assume that its training data and process are very similar to Gemini-1.5-Pro.
> - **AlanVLM**: We have included AlanVLM as an extra baseline (the LMM trained on EVUD), serving as a supplementary baseline for synthetic data.
> - **New benchmarks:** We have added results from the HD-EPIC VQA benchmark and the open-ended OpenEQA.  We also convert the multi-instance retrieval (V → T) task in Epic Kitchens 100 into a multiple-choice task and compared the mAP performance with EgoVLP and LaViLa-L.
>
> |                        | EgoSchema   | EgoPlan     | QAEgo4D         | EgoTaskVQA  | OMEBench (beh.)  | OMEBench (vis.)  | HD-Epic        | OpenEQA    | EP-100 MIR(V →T) |
> | ---------------------- | ----------- | ----------- | --------------- | ----------- | ---------------- | ---------------- | -------------- | ---------- | ---------------- |
> | EgoVLP                 | -           | -           | -               | -           | -                | -                | -              | -          | 26.0             |
> | LaViLa-L               | -           | -           | -               | -           | -                | -                | -              | -          | 40.0             |
> | Gemini-2.0-Flash       | **71.1**    | **51.5**    | 62.4            | **59.2**    | 60.2             | 62.1             | 38.8           | **61.5**   | 49.3             |
> | AlanVLM                | 50.7        | 27.6        | 41.8            | 43.2        | 27.3             | 33.2             | 25.7           | 46.7       | 23.1             |
> | Qwen2.5-VL             | 65.0        | 45.2        | 59.4            | 53.7        | 55.3             | 49.4             | 34.2           | 55.7       | 42.1             |
> | VideoChat-Flash        | 64.3        | 40.8      | 62.4            | 54.8        | 57.1             | 54.5             | 39.4           | 58.1       | 45.3             |
> | OpenMMEgo（Qwen2.5-VL）  | 69.3 (+4.3) | 50.2 (+5.0) | **65.6** (+6.2) | 56.2 (+2.5) | **65.7** (+10.4) | **63.2** (+13.8) | **42.4**(+8.2) | 60.2(+4.5) | **49.7** (+7.6)  |
>
> Furthermore, the reviewer notes that certain prior works in token compression, synthetic data, and curriculum learning could serve as alternatives to some of our components, thus forming potential baselines. Among the potential baselines pointed out by the reviewer, due to time and resource constraints, we adapt VTM  to OpenMMEgo as the baseline for video token compression and the strategy from short to long videos as the baseline for curriculum learning. In terms of synthetic annotations, LLaVA-Video is already our baseline, and we have additionally included a comparison with AlanVLM (fine-tuned on synthetic data EVDU), so no further comparison is made. We perform SFT on Qwen2.5-VL with 1.6M data.
> |                        | EgoSchema   | EgoPlan     | QAEgo4D         | EgoTaskVQA  | OMEBench (beh.)  | OMEBench (vis.)  | HD-Epic        | OpenEQA    | EP-100 MIR(V →T) |
> | ---------------------- | ----------- | ----------- | --------------- | ----------- | ---------------- | ---------------- | -------------- | ---------- | ---------------- |
> | OpenMMEgo              | 66.5        | 47.7        | 62.6            | 53.9        | 62.1             | 58.0             | 36.1           | 58.4       | 46.1             |
> | OpenMMEgo w. VTM       | 65.8        | 46.3        | 61.8            | 52.1        | 60.4             | 55.7             | 34.2           | 54.3       | 45.2             |
> | OpenMMEgo w. shot→long | 65.3        | 45.9        | 62.4            | 53.7        | 58.7             | 56.2             | 35.4           | 53.1       | 45.7             |
>
> Aside from being comparable to Gemini 2.0 on certain metrics, OpenMMEgo consistently outperforms all other baselines, further demonstrating the effectiveness of our method.
>
> ---
>
> **Token Compression Cost vs. Benefit**
> Token compression does incur additional computation. Normalizing the inference time of “w/o STM+TTP” to 1.0, we compare supported maximum frames and relative cost:
>
>
> |                | w/o STM+TTP | w/o STM | w/o TTP | OpenMMEgo |
> | -------------- | ----------- | ------- | ------- | --------- |
> | frames         | 64          | 106     | 91      | 192       |
> | inference cost | 1.0         | 1.31    | 1.19    | 1.47      |
>
> Our token compression strategy enables processing **two times additional frames** with only a 47 % increase in inference time.
>
> **Missing Details / Clarifications**
> We appreciate the reviewer’s corrections and provide the following clarifications:
> 1. **Figure 5:** All answers are produced by OpenMMEgo; “EgoCatch” was a mislabel.
> 2. **Figure 3 (right):** Our radar chart is plotted with the scale set to 0 and 1, where the minimum value is a multiple of 5 below the lower limit and the maximum value is used as the upper limit. However, for aesthetic purposes, we manually labeled the values in the plot with PowerPoint. Unfortunately, due to a saving error, there are two typos (OpenMMEgo w/o STM+TTP's Gen.vis. is 67.5, and OpenMMEgo's Egoplan is 47.7). The proportions of the plot itself are correct. We sincerely apologize for any confusion caused.
> 3. **L171–172 (OMEBench Verification):** No, limited by our human resources.
> 4. **L203 (Merging):** We merge tokens within each component by summing, following ToME.
> 5. **L215 (“Excessive Semantic Similarity”):** Tokens with excessive semantic similarity are pruned only in fast frames. On one hand, in fast frames, we focus on the dynamic changes of these semantic tokens. In our visualization (Fig. 4), the tokens with the highest similarity are often static ones, such as background elements, while the dynamic parts are not pruned. On the other hand, in slow frames, the static temporal information of these semantic tokens is preserved.
> 6. **L245–247 (Curriculum Difficulty):** In the Offline Data Curriculum, we are more concerned with the inherent difficulty of the data itself rather than its ease of learning. Using a fine-tuned model to evaluate might be influenced by the extent to which the data can be optimized. In the Online Data Curriculum, our dropout mechanism compensates for this.
> 7. **Figure 3 (left):** We will update the left plot per the suggestion; the right-plot typos have been corrected as above.
>
> We hope these clarifications resolve the reviewer's concerns. Please let us know if any further questions.

---

> > ### Author Response · Authors · 2025-08-05
> >
> > We are reaching out to follow up on our rebuttal. We would very appreciate any feedback or clarification on whether our responses addressed the reviewer's concerns. The reviewer’s insights and expertise are highly valued, and any additional comments would be tremendously helpful for us to better refine and strengthen the work. We truly appreciate the time and thoughtful effort the reviewer has invested in evaluating our submission.

---

> ### Comment · Reviewer_iYek · 2025-08-06
> **Reviewer response to rebuttal from authors**
>
> I thank the authors for their response and additional experiments. In particular, I appreciate the comparisons on new benchmarks, curriculum learning with short -> long videos, and token compression cost analyses. I also appreciate the clarification about the radar plot. These addressed some of the specific concerns I had. However, I'm not convinced about the token compression, synthetic annotations and some baseline comparisons (detailed below). However, since the majority of concerns were addressed, I'm raising my rating.
>
> * **Token compression:** This is one of the core contributions (L56 - 62). The paper currently lacks a discussion of what prior token compression methods do and how this paper innovates over it. Simply citing three (among a large list) of past work without any discussion or experimental comparison is insufficient. That said, I thank the authors for the new results with VTM. While this is only a single data point (given the rich literature of token merging), this is useful. Is this with the Learnable VTM approach here in Figure 4: https://arxiv.org/pdf/2410.23782?
> * **Synthetic annotations:** LlavaVideo dataset is a baseline, but the approach proposed for generating the LlavaVideo dataset is not compared against. I would like to emphasize this difference. Simply using prior datasets in comparison is not what I'm suggesting. My key concern is that the *data curation* approach proposed here is not vetted against prior data curation approaches in an apples-to-apples fashion:
> 	* The current experiments do not clearly demonstrate the importance of the technical contributions in the data generation pipeline. This needs to be done on the same videos with the same frontier models for generation.
> 	* "core visual fact" extraction  (L137), multi-step prompting (L140), and generating QAs for different time scales (L153) are different in comparison to prior work. How much of a role do these play in comparison to techniques used in LlavaVideo?
> 	* Furthermore, LlavaVideo uses GPT-4o whereas the proposed method uses Gemini-1.5 Pro. For example, GPT-4o (71.9) is worse than Gemini 1.5 Pro on VideoMME (73.2). LlavaVideo also uses a different collection of videos than OpenMMEgo. Purely comparing the datasets (and not the methodology) can be misleading.
> * **Baseline comparison is not apples-to-apples:**
> 	* **For context on Gemini 2.0 Flash vs. Gemini 1.5 Pro:** Gemini 1.5 Pro is quite a bit better on video benchmarks as shown in Table 6 here: https://storage.googleapis.com/deepmind-media/gemini/gemini_v2_5_report.pdf. So the numbers reported here are likely worse than what 1.5 Pro would have scored.

---

> ### Author Response · Authors · 2025-08-07
> **Response to Reviewer Comments in the Discussion Stage (part1)**
>
> We sincerely thank the reviewer for raising the score and for acknowledging that many of the key concerns were addressed. We appreciate the positive feedback on our new benchmark comparisons, curriculum learning design, token compression analysis, and the clarification regarding the radar plot.
> We also thank the reviewer for clearly identifying the remaining concerns, which allows us to provide focused responses. Should any point require further clarification, we would be pleased to provide it.
>
> ---
> **Token Compression**:
>
> Thanks for the valuable suggestion regarding the discussion. We recognize that our initial approach—briefly introducing the background in the introduction (L55-62) while focusing on VLM-specific compression in related works (L96)—may have obscured the clarity of our contributions. We appreciate the reviewer for highlighting this issue and providing relevant references on token merging. Since NeurIPS does not permit uploading a revised manuscript, we write a draft below, which **will be added in the next version as promised**:
>
> >Visual token merging has demonstrated significant effectiveness in improving representation efficiency while reducing computational costs, initially in image classification tasks. Key techniques include similarity-based merging (ToMe)[1], learned merge ratio (Token Merger)[2], learned threshold-based merging and pruning (LTMP)[3], decoupled embedding modules (DTEM)[4], and depth-aware spatial token aggregation (ToSA)[5]. These approaches have since been extended to video inputs, with innovations such as learnable spatiotemporal merging (VTM) [6], spatial-temporal token selection (STTS)[7], and temporal interpolation (InTI) [8]. Similarly, in large multimodal models (LMMs), token compression has been adopted to improve inference efficiency and handle long contexts, including methods based on static pixel token reduction (Video-LLaMA3)[9], cross-modal querying reduction (LongVU)[10], and hierarchical compression (VideoChat-Flash)[11]. Building on these advances, we propose a method specifically designed for egocentric videos, which present unique challenges due to dense visual cues and frequent camera motion. We introduce a dual-branch compression strategy that restricts spatial token merging to adjacent tokens for preserving semantic entity boundaries, and performs temporal pruning over entity-aligned tokens in a SlowFast-inspired view to retain motion-salient information.
>
> Regarding the specific experimental point: Yes, we implemented the learnable VTM variant in our framework exactly as depicted in Figure 4 of the paper (https://arxiv.org/pdf/2410.23782). While we acknowledge that including a broader set of baselines would have strengthened our evaluation, we were constrained by the limited time and computational resources available during the rebuttal period. We are particularly grateful that the reviewer recognizes the value of our OpenMMEgo with VTM as a ‘useful' contribution to the experimental analysis.
>
> [1] Token Merging: Your ViT but Faster.
>
> [2] Efficient Vision Transformer via Token Merger.
>
> [3] Learned Thresholds Token Merging and Pruning for Vision Transformers.
>
> [4] Learning to Merge Tokens via Decoupled Embedding for Efficient Vision Transformers.
>
> [5] ToSA: Token Merging with Spatial Awareness.
>
> [6] Video Token Merging for Long-form Video Understanding.
>
> [7] Efficient Video Transformers with Spatial-Temporal Token Selection.
>
> [8] Dynamic and Compressive Adaptation of Transformers From Images to Videos.
>
> [9] VideoLLaMA 3: Frontier Multimodal Foundation Models for Image and Video Understanding.
>
> [10] LongVU: Spatiotemporal Adaptive Compression for Long Video-Language Understanding.
>
> [11] VideoChat-Flash: Hierarchical Compression for Long-Context Video Modeling.

---

> ### Author Response · Authors · 2025-08-07
> **Response to Reviewer Comments in the Discussion Stage (part2)**
>
> **Data Comparison**:
> 1. We fully appreciate the reviewer's suggestion that apples-to-apples comparisons of data curation pipelines using the same video set and model would be insightful. However, at our current dataset scale, re-annotating the entire corpus with multiple alternative construction methods is unfortunately infeasible due to the high cost of high-quality video-language annotation. This cost consideration also motivated our choice of Gemini 1.5 Pro over GPT-4o, as it provides more economical scaling while maintaining quality.
> 2. Instead, our current focus has been on two complementary aspects: (a) Demonstrating the value of our dataset relative to existing baselines; (b) Validating that each of the two compositions, particularly the behaviour-based and vision-centric data, contributes positively to the performance of downstream models.
> 3. The core contribution of our paper is introducing a scalable, practical data construction pipeline for egocentric video understanding. While comparative studies of annotation strategies would indeed be valuable, such analysis is beyond the scope of this work. Our current focus is to clearly document our pipeline's construction and demonstrate its effectiveness through measurable performance gains.
>
> ---
> **Baseline Comparison**:
>
> We appreciate the reviewer’s question regarding model comparison fairness. Our selection of **Gemini 2.0 Flash** over **Gemini 1.5 Pro** was carefully considered based on the three key factors:
> 1. **Availability**: Gemini 1.5 Pro was no longer accessible through official API channels during our evaluation period.
> 2. **Model size and usage context**: Gemini 1.5 Pro is a significantly larger model and not aligned in scale with our 7B-size model. Gemini 2.0 Flash provides a more appropriate comparison point, being both closer in size and trained on similar corpora.
>
> More importantly, **Gemini  2.0 Flash can outperform Gemini 1.5 Pro on some egocentric video benchmarks**, which is contrary to our common expectation based on the Gemini 2.5 technical report. To prove it, we provide two examples:
> - On the **EgoSchema**, Gemini 1.5 Pro scores **63.5** (Table 10 in https://arxiv.org/pdf/2403.05530v2), while Gemini 2.0 Flash scores **71.1**.
> - On the **EgoTempo**, Gemini 1.5 Pro scores **36.3**, whereas Gemini 2.0 Flash achieves **39.3**. (Table 6 in https://storage.googleapis.com/deepmind-media/gemini/gemini_v2_5_report.pdf)
>
> Therefore, we believe the use of Gemini 2.0 Flash actually represents a **stronger** baseline for egocentric video understanding, rather than a weaker one. We will clarify this choice more explicitly in the revised draft.
>
> ---
>
> We sincerely thank the reviewer once again for the insightful suggestions. We greatly appreciate the time and effort spent on reviewing our work. We will revise the paper to better reflect the points raised and to improve the overall clarity and completeness.

---

### Decision · Program_Chairs · 2025-09-17

**Decision:**

Accept (poster)

**Comment:**

The paper receives consistent positive ratings from the reviewers. They appreciate the novelty of the proposed token reduction strategy and the curated large-scale benchmark OEM-10k for ego-centric video understanding with VLM. The reviewers had questions about some details of the method design and experiments. The authors' rebuttal addressed the problems well. AC also agrees with the comments of the reviewers and recommends acceptance as a poster for this submission.